



# Sources and pathways of biocides and their transformation products in urban water infrastructure of a 2ha urban district

Felicia Linke[1, 3], Oliver Olsson[2], Frank Preusser[3], Klaus Kümmerer[2], Lena Schnarr[2], Marcus Bork[1], Jens Lange[1]

[1]Hydrology, Faculty of Environment and Natural Resources, University of Freiburg, 79098 Freiburg, Germany
[2]Institute of Sustainable and Environmental Chemistry, Leuphana University of Lüneburg, 21335 Lüneburg, Germany
[3]Institute of Earth and Environmental Sciences, University of Freiburg, 79104 Freiburg, Germany

*Correspondence to*: Felicia Linke (felicia.linke@hydrology.uni-freiburg.de)

**Abstract.** Biocides used in film protection products leaching from facades are known to be a potential threat for the environment. This study identifies individual sources and entry pathways in a small-scale urban area. We investigate emissions of commonly used biocides (terbutryn, diuron and octylisothiazolinone (OIT)) and some of their transformation products (TPs; diuron-desmethyl, terbumeton, terbuthylazin-2-hydroxy and terbutryn-desethyl) from a 2 ha residential area, 13 years after construction has ended. Sampling utilizes existing urban water infrastructure representative for decentralized storm water management in central and northern Europe and applies a two-step approach to (a) determine the occurrence of biocides above water quality limits (i.e. predicted no effect concentration, PNEC) and (b) identify source areas and characterize entry pathways into surface- and groundwater. Monitoring focuses on the analysis of selected biocides and TPs by LC-MS/MS in water samples taken from facades, rainwater pipes, drainage and storm water infiltration systems. In standing water in a swale we found high concentrations of diuron (174 ng L$^{-1}$) and terbutryn (40 ng L$^{-1}$) above PNEC for surface water. We confirmed expected sources, i.e. facades, but sampling of rain downpipes from flat roofs identified additional sources of all biocides and two TPs of terbutryn and one TP of diuron. Diuron and terbutryn were found in three drainage pipes representing different entry pathways of biocides. In one drainage pipe collecting road runoff only diuron-desmethyl and terbutryn-desethyl were detected. In two other drainage pipes collecting infiltrated water through soil additionally terbuthylazin-2-hydroxy was detected. Concentration of terbutryn and two of its TPs (terbutryn-desethyl and terbuthylazin-2-hydroxy) were highest in one of the pipes collecting infiltrated water through soil which suggests a high leaching potential of this biocide. This study shows that target-oriented monitoring of urban water infrastructure for biocides and their TPs allows for a better identification of biocide emissions into urban aquatic environments.

## 1 Introduction

Biocides are bio active substances, and regulated by the EU Regulation 528/2012 (European Parliament and Council, 2012). They are, for example, used in renders and paints to inhibit the growth of microorganisms (e.g. algae and fungi) on facades (Sauer, 2017). The same active substances used as pesticides in agriculture can act as biocides in urban areas. In agriculture,



the environmental fate of selected pesticides has been investigated since first usage of e.g. diuron (Peck et al., 1980; Giacomazzi and Cochet, 2004) and terbutryn (Wu et al., 1974; Donati and Funari, 1993; Musgrave et al., 2011). Pesticides in agriculture have intensively been characterized regarding their sources (Reichenberger et al., 2007), environmental compartments for transformation (Fenner et al., 2013; Gassmann et al., 2013) and pathways to surface- and underground waters

(Doppler et al., 2012; Greiwe et al., 2021). However, this is not the case for biocides applied in the urban context. Diverse applications result in a variety of different sources and entry pathways (Burkhardt et al., 2011; Wieck et al., 2018; Paijens et al., 2021). Environmental impacts are relevant, since in catchments with mixed land use, overall loads of urban biocides were found to exceed those of agricultural pesticides (Wittmer et al., 2010).

Terbutryn, diuron and octylisothiazolinone (OIT) represent commonly used biocides in coatings. To date, all three biocides

are transitionally authorized as film and construction materials preservatives, i.e. Product-Type 07 and 10, in the EU. Use of terbutryn and OIT is additionally legal for fibre, leather, rubber and polymerized materials preservatives (Product-Type 09) and OIT additionally for products from preservatives for products during storage (Product-Type 06), wood preservatives (Product-Type 08), preservatives for liquid-cooling and processing systems (Product-Type 11) and working or cutting fluid preservatives (Product-Type 13) (ECHA, 2007-2020). Yet, diuron and terbutryn are regulated in the Water Framework

Directive as prioritized substances (European Parliament and Council, 2013) and for final authorization as biocides, there is an ongoing risk assessment until 2024. Predicted no effect concentration (PNEC) values in fresh water are 0.02 µg L$^{-1}$ for diuron, 0.013 µg L$^{-1}$ for OIT and 0.034 µg L$^{-1}$ for terbutryn (Paijens et al., 2019).

Transformation of biocides can principally occur directly on treated objects (e.g. by photolysis on facades, Hensen et al., 2018) or along environmental pathways (e.g. in the soil, Bollmann et al., 2017a.). Degradation time of terbutryn in soil ranges

between 10 days (Lechón et al., 1997) and 231 days (Bollmann et al., 2016) depending on, for example, temperature, pH, organic and clay content. Terbutryn half-life in water under aerobic and anaerobic conditions were reported to be 193 - 644 days and 266 - 400 days, respectively (Talja et al., 2008). Diuron is highly persistent in soil, sediments and water. It is slowly degraded by hydrolysis and biodegradation with a half-life of a month up to a year (Giacomazzi and Cochet, 2004). Johann et al. (2018) showed an increase of sorption of diuron in the soil passage with an increase of organic matter. Bollmann et al.

(2017a) estimated a half-life of diuron of more than 2500 days in soil. OIT has a reported half-life of 9.3 days (Bollmann et al., 2017b).

Diuron, terbutryn and OIT degrade to various transformation products, TPs (Hensen et al., 2020). In this study we focus on three TPs of terbutryn (terbuthylazin-2-hydroxy, terbutryn-desethyl, terbumeton) and one TP of diuron (diuron-desmethyl). Terbuthylazin-2-hydroxy and terbutryn-desethyl are formed by photolysis or biodegradation (Bollmann et al., 2017a; Hensen

et al., 2018; Burkhardt et al., 2012; Bollmann et al., 2016). In a leaching study under natural weather conditions, Bollmann et al. (2016) found terbuthylazin-2-hydroxy, terbutryn-desethyl and terbumeton in render and in leachate. Terbumeton is a photo degradation product that tends to remain on facades (Bollmann et al., 2017a). Terbutryn-desethyl, terbuthylazin-2-hydroxy and terbumeton were classified as probably toxic (Hensen et al., 2020). Diuron-desmethyl was identified as a photo TP (Burkhardt et al., 2012; Hensen et al., 2018) and but is possibly also formed by microorganisms in soil (Hensen et al., 2018).





Diuron-desmethyl was detected in urban runoff by various studies (Wittmer et al., 2010; Reemtsma et al., 2013; Hensen et al., 2018). In a field experiment only 0.4 % of the diuron losses were made of diuron-desmethyl (Burkhardt et al., 2012). Moschet et al. (2014) confirmed diuron-desmethyl in rivers in Switzerland at concentrations ranging from 10 to 22 ng L$^{-1}$. Diuron-desmethyl was classified as most probably toxic or probably toxic (Hensen et al., 2020).

Biocides and their TPs are washed off from facades and enter the environment during rain events. In urban areas, diuron,
terbutryn, OIT and their known TPs were detected in storm water (Burkhardt et al., 2011), surface water (Quednow and Püttmann, 2007), waste water treatment plants (Bollmann et al., 2014a), soil (Bollmann et al., 2017a) and groundwater (Hensen et al., 2018). This demonstrates the need to further understand sources, transformation and pathways of these biocides applied on facades based on substance behavior. Various laboratory studies on leaching of film preservatives of facades exist (Jungnickel et al., 2008; Schoknecht et al., 2009; Wangler et al., 2012; Schoknecht et al., 2013; Bergek et al., 2014; Styszko
et al., 2015; Urbanczyk et al., 2016) complemented by experimental studies under natural weather conditions (Burkhardt et al., 2012; Bollmann et al., 2016; Schoknecht et al., 2016; Bollmann et al., 2017a; Vega-Garcia et al., 2020). Release of biocides from facades is controlled by temperature, time between rain events, their extent, wind, UV exposure, biocide characteristics and properties of paint and renders used (Paijens et al., 2019), as well as architectural design.

Compared to experimental investigations, studies monitoring biocidal chemicals in urban storm water systems are relatively
rare. They mostly focus on large heterogeneous areas where individual sources and entry pathways can hardly be identified (Wicke et al., 2015; Paijens et al., 2021). Studies confirmed general biocide emissions from larger heterogeneous residential areas in storm water channels of separated sewer systems (Bollmann et al., 2014b; Wicke et al., 2015). In the scale of urban districts, monitoring so far concentrated on newly built areas to identify maximum biocide loads, shortly after construction had been finished (Burkhardt et al., 2011; Bollmann et al., 2016). In another study, Hensen et al. (2018) investigated biocide
emission from two small urban catchments with sizes of 2.95 ha and 8,047 m², but focused on the receiving parts of the water infrastructure, a swale-trench system.

Much is still unknown on the emission potential and related environmental risks of biocides used in real-word urban areas. This is mainly due to the fact that existing studies do not systematically follow the fate of biocides including their TPs from source to sink in urban districts with buildings' ages of 10 years or more. The aim of this study is therefore to identify sources
and pathways of terbutryn, diuron and OIT used for film protection in a 2 ha urban residential area, 13 years after initial painting, and to monitor some of their known TPs (diuron-desmethyl, terbuthylazin-2-hydroxy, terbutryn-desethyl and terbumeton). These TPs have been already investigated in previous studies about biocide emissions from facades (Burkhardt et al., 2012; Bollmann et al., 2016) and analytical standards are available, thus quantification is possible.

Using a stepwise approach and making use of existing urban water infrastructure, this study characterizes the environmental
hazard of urban biocide use with only a small number of samples (n= 52), thus limiting laboratory expenses. The main aim is to identify starting points and pathways where the release and transport of substances into adjacent aquatic systems occurs. Thereby, potential sources of biocides and TPs are identified by sampling at sselected points of urban infrastructure, e.g. rain downpipes of flat roofs and by artificial experiments on facades and roof materials. After the identification of main emission





sources, pathways of biocides and TPs from the source to the catchment outlet are analyzed by sampling at selected drainage
pipes or storm water canals. This also allows a comparison of the different pathways with regard to their degradation potential.

## 2 Methods

### 2.1 Two-step approach

We developed a two-step approach utilizing existing urban water infrastructure (Fig. 1). This was done to first check the
relevance of biocide emission in the study area and then investigate sources and entry pathways. In a first step, we verified the
occurrence of terbutryn, diuron and OIT and their TPs (diuron-desmethyl, terbuthylazin-2-hydroxy, terbutryn-desethyl and
terbumeton) in urban aquatic systems and therefore the potential of biocide leaching from building materials by sampling in a
selected urban water infrastructure at the outlet of the study area. If the determined concentrations were higher than the
available guidance values (e.g. Measured Environmental Concentrations/ Predicted no Effect Concentrations (MEC/PNEC)
>1) there is a risk and thus a relevance to take further measures for risk mitigation. In this study, MEC/PNEC was chosen. If
this criterion is fulfilled, Step 2 is carried out to identify sources and pathways of selected biocides and their TPs. Finally, this
knowledge may guide measures to mitigate biocide pollution.

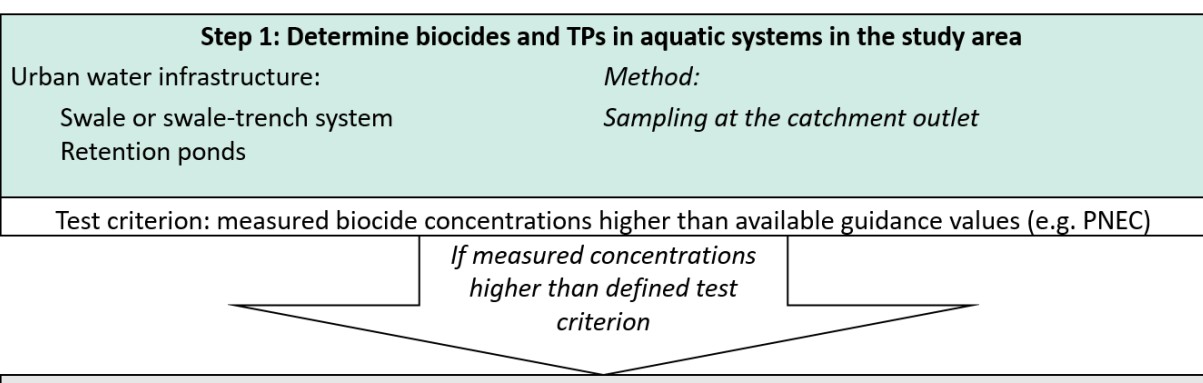

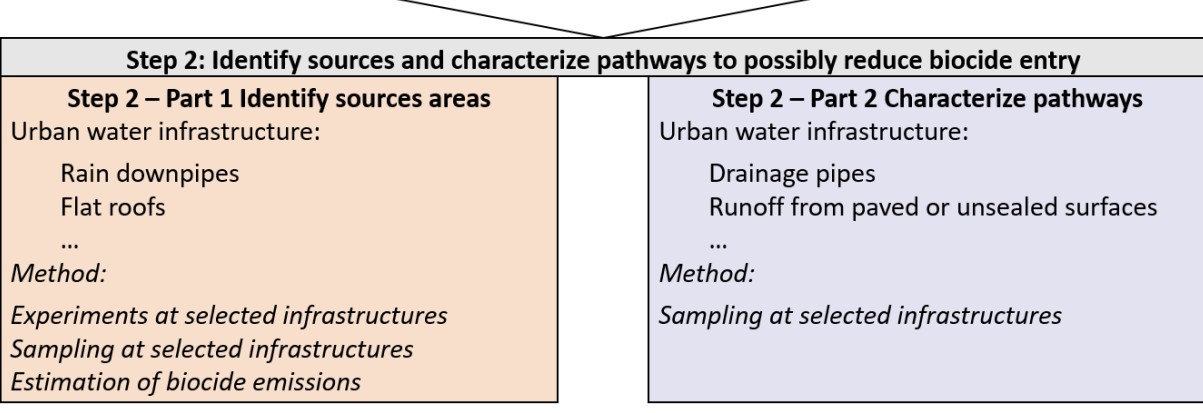

**Figure 1: Two-step approach focusing on urban water infrastructure.**





## 2.2 Study Area and sampling sites

The study area is located in the city of Freiburg in south-western Germany. It comprises a residential district of approximately 2 ha with buildings of uniform construction time, type and structure. The modern four-story houses with thermal insulation composite systems were lastly painted in 2007 according to a survey among residents and architects. The development plan was fixed in 2005 and construction started in 2006 (Stadt Freiburg i. Br., 2005). Roof areas are of diverse use such as roof top terraces and solar panels, both in combination with extensive green roofs. Houses have no roof overhang. Similar development

areas exist in other parts of the city and are typical for modern architecture in northern and central Europe.

The study area consists of eight houses connected to a separated sewer system that ends up in a swale (Fig. 2). The focus of the sampling campaign was on four houses (1, 2, 4 and 5) and on surface runoff from a neighboring street that accepts water from three additional houses (6, 7 and 8). All houses were constructed at the same time and thus exposed to identical weather conditions over the years. At two houses (3, 4) used render contains diuron and OIT according to inhabitants and invoices of

construction work. For the other houses used paint or renders could not be identified.

Houses 1, 2 and 4 all have flat roofs that are mostly covered by extensive greening. House 1 additionally has small roof top terraces. House 2 has an extensive green area with solar panels but no roof top terrace. House 4 contains two larger roof top terraces.

We placed our sampling sites along assumed biocide source areas and pathways (Fig. 3). Names of sampling points correspond

to house numbers. Assumed sources were **facades** of buildings (**F**) and roof areas. Storm water from roof areas drains into rain downpipes. These rain downpipes were sampled at three individual houses (**roof runoff sampling points, R**). Water from rainwater downpipes flows via paved gutters into a small grass-covered trench along the houses. This trench leads into the swale system. Surface runoff of a neighboring street is diverted by an underground pipe and drains into the same trench. Water samples were taken at this pipe (**surface water pipe, S**).





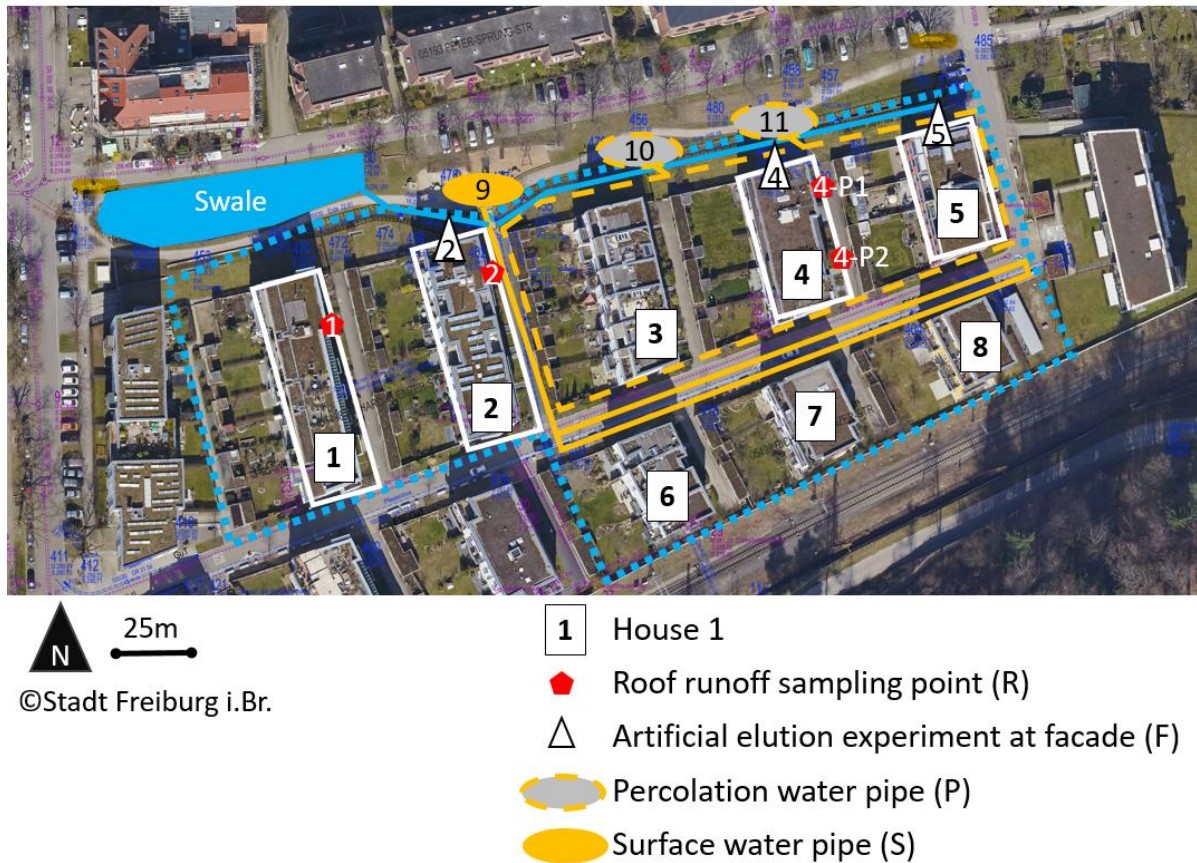

**Figure 2: Study area with sampling points and houses, aerial view. White lines indicate investigated houses. The blue dotted line encloses the drainage area connected to the swale. Yellow and dashed yellow lines enclose approximate areas connected to percolation water pipes and surface water pipe. Source: Stadt Freiburg i.Br.**

A special feature of the study area is the possibility to collect water samples after soil passage. A drainage system above an

underground parking garage located beneath houses 3 to 5 collects all water infiltrating from gardens and green areas surrounding the houses including facade runoff. The soil consists of a 10 - 20 cm topsoil layer covered with grass. Beneath, the soil is composed of expanded clay aggregates to reduce the weight on the underground parking garage. At regular distances, there are pipe outlets of this drainage system directing water to the small trench at the northern site of the study area. Two of these pipes, hereafter referred to as **percolation water pipes (P)**, were selected for sampling, P10 and P11. Both pipes represent

the surrounding area of a house with all the infiltrating water.





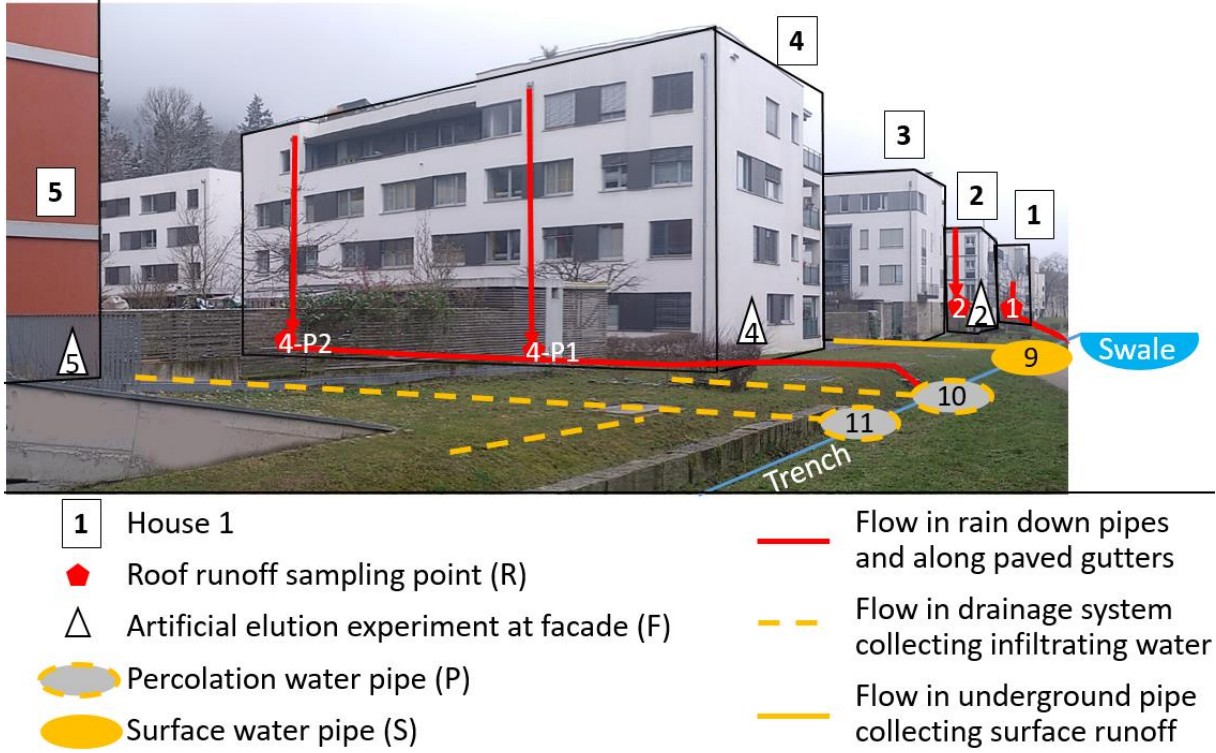

**Figure 3: Side view of storm water management infrastructure and sampling points.**

## 2.3 Sampling

Water samples were collected in 1L brown glass bottles previously washed with deionized water. All samples were
immediately stored at 4ºC and sent to the laboratory within 24 hours after sampling.

### 2.3.1 Event samples

From 2015 to 2017, we took samples from the swale system (Step 1, Figure 1) during four individual rain events (Table 1).
Samples were taken as grab samples during the events. In 2019 to 2020 roof rainwater was sampled at rain downpipes from
houses 1, 2 and 4 to account for different roof area usages (Step 2, Part 1). For houses 1 and 2 samples were taken at one pipe,
for house 4 at two pipes. A second pipe at house 4 was additionally sampled, since the first pipe showed high biocide
concentrations during initial sampling. Duplicate samples were taken at least during one event for each pipe. For Step 2 (Part
2), water samples were collected at two percolation water pipes (P10, P11) and at the pipe collecting surface water from a
neighboring street (S9). We sampled all three pipes during four events including duplicate samples during the first two events.






**Table 1: Overview of samples taken during rain events.**

| Step | Sample Category | Name | Samples taken per event (Duplicates >1) | | | | | | # Events sampled |
|------|-----------------|------|------|------|------|------|------|------|------|
| 1 | Swale | | 2015-11-20 | 2016-02-09 | 2016-11-06 | 2017-05-19 | | | |
| | | Swale | 1 | 1 | 2 | 1 | | | 4 |
| 2 Part 1 | Rain downpipes | | 2019-11-29 | 2020-02-11 | 2020-03-02 | 2020-03-10 | 2020-05-11 | 2020-06-28 | |
| | | R1 | 0 | 0 | 0 | 1 | 2 | 2 | 3 |
| | | R2 | 0 | 1 | 0 | 1 | 2 | 1 | 4 |
| | | R4-P1 | 2 | 2 | 2 | 1 | 0 | 2 | 5 |
| | | R4-P2 | 0 | 2 | 2 | 1 | 0 | 2 | 4 |
| 2 Part 2 | Drainage Pipes | | 2019-07-28 | 2019-09-08 | 2019-11-29 | 2020-02-11 | 2020-03-10 | | |
| | | S9 | 0 | 2 | 2 | 1 | 1 | | 4 |
| | | P10 | 2 | 0 | 2 | 1 | 1 | | 4 |
| | | P11 | 2 | 0 | 2 | 1 | 1 | | 4 |

Fig. 4 shows daily precipitation at a climate station about 5 km away from the study area and the time of sampling in swale, at drainage pipes and at rain downpipes. Daily rainfall ranged between 0 mm and 54.6 mm. The first swale sample was collected

shortly after the highest rainfall in the six-year monitoring period.



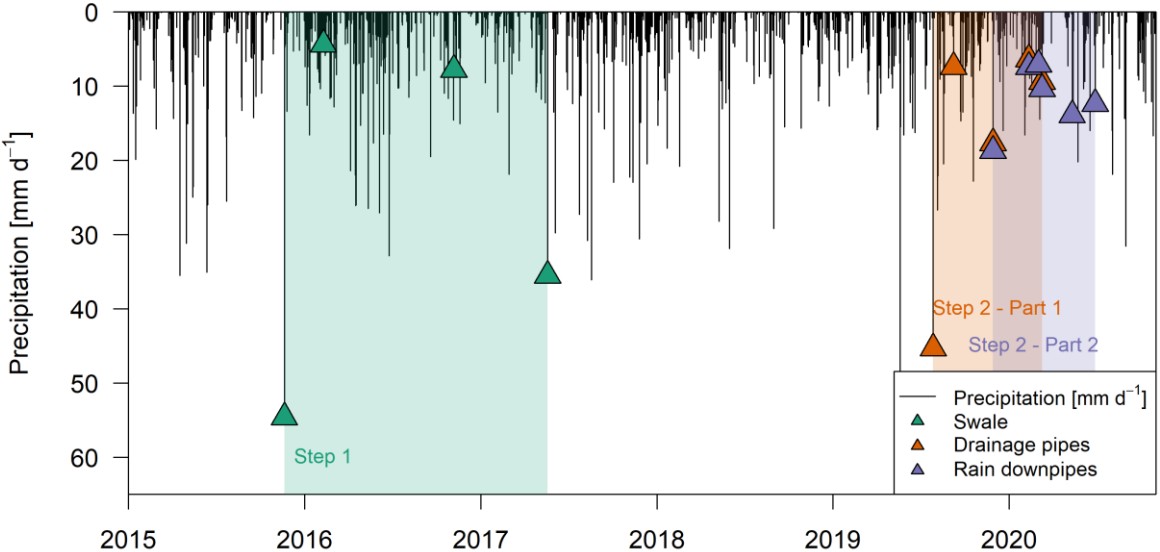

**Figure 4: Daily precipitation about 5 km away from study site between 2015 and 2020. Sampled events are marked according to the two-step approach. Precipitation data taken from Deutscher Wetterdienst, Station Freiburg.**

**2.3.2 Elution experiments**

To determine potential biocide wash off and thus significance of different sources of biocides, we conducted artificial elution experiments. We sprinkled facades with 1L of deionized water. The water was poured within 30 s with a measuring cup across an area of approximate 0.25 m² of the facade. We collected the water flowing down the facade using a rectangular container made out of stainless steel held against the facade. The container was cleaned with acetone prior to its use. The collected water

was stored in brown glass bottles. Water samples were collected at the facade oriented northwest at houses 2, 4 and 5. Additional, several elution experiments were conducted on the roof area of house 4. They included roof facades (n=6) and roof materials, namely wooden terraces (n=6), railings (n=2), roofing foils (n=2), roof access (n=1), roof cladding (n=1), elevator shaft foil (n=1), and grass foil (n=1).

**2.3.3 Leaching test**

An additional leaching test was performed on the wooden terrace. A wooden part of the terrace was removed and sawn into two pieces each with a volume of 128 cm³. Each piece was put into 500 mL of deionized water and shaken for 24 h. Then, the water was changed and shaken again for 24 h.





### 2.4 Chemical analysis

#### 2.4.1 Chemicals and reagents

Analytical standards of diuron, diuron-desmethyl, terbutryn, terbutryn-d5, terbumeton, terbuthylazin-2-hydroxy and terbutryn-desethyl were purchased from Neochema (Bodenheim, Germany). Diuron-d6 was received from hpc-standards (Borsdorf, Germany). OIT was purchased from LGC-Standards (Teddington, UK). Acetonitrile (LCMS grade; VWR International GmbH, Darmstadt, Germany) was used as organic mobile phase in chromatography and for the preparation of stock solutions.

#### 2.4.2 Preparation and measurement of environmental samples

Environmental samples (0.9 L for surface water) were filtered with a folded filter (type 113 P Cellulose ø 240 mm, Carl Roth GmbH + Co. KG, Germany). The filtrate was spiked with the internal standard diuron-D6 and terbutryn-D5 (20 µl of 1 mg L$^{-1}$, respectively). For solid phase extraction (SPE), cartridges (CHROMABOND® HR-X 6 mL 200 mg$^{-1}$) were conditioned with 10 mL methanol and washed with 10 mL pure water. Environmental samples were enriched on the cartridges via Teflon capillary and a vacuum extraction unit. After enrichment of the samples, cartridges were washed with 5 mL pure water and air

dried for 5 - 10 minutes. Elution was done with 10 mL of a mixture of methanol and chloroform (v/v; 1:1). The eluted phase was dried with nitrogen and resolved in 200 µl acetonitrile. Measurements of environmental samples were performed with a Triple Quadrupole (Agilent Technologies, 1200 Infinity LC-System and 6430 Triple Quad, Waldbronn, Germany) with ESI in positive mode. A C18 column (Nucleodur 100-3 C18ec, 3 µm particle size, 125 mm length, 2 mm diameter from Macherey & Nagel) was used as stationary phase and temperature set to 30 °C. Acetonitrile (A) and water with 0.1% formic acid (B)

were used as mobile phase. Gradient was 0 - 1min (75 % B), 1 - 7 min (40 % B), 7 - 12 min (15 % B), 12 - 15 (15 % B) 15 - 17 min (75 % B), 17 - 20 (75 % B). Flow was 0.4 mL min$^{-1}$ and the injection volume 5 µL.

Biocides and their TPs were identified and quantified on the basis of a precursor and two fragment ions (quantifier and qualifier). During each analysis, calibration standards covering 0.5 - 1000 µg L$^{-1}$ were measured. For all substances except terbuthylazin-2-hydroxy, the last four calibration standards (100, 250, 500 and 1000 µg L$^{-1}$) were only taken into account if

enriched sample concentrations exceeded 50 µg L$^{-1}$. Samples with biocides concentrations > 1000 µg L$^{-1}$ (after enrichment) were diluted and extraction and measurement were repeated.

Concentration of terbutryn, its TPs as well as OIT were calculated in reference to terbutryn-d5. Concentrations of diuron and its TP were calculated in reference to diuron-D6. Analysis was performed with the MassHunter software *QQQ Quantitative Analysis (Agilent Technologies)*.

Pure water as a blank sample and a reference with 100 µg L$^{-1}$ terbutryn, diuron and OIT were carried along each analysis as quality control.

Limits of detection (LOD) and quantitation (LOQ) were calculated with DINTEST (2003) according to DIN 32645 (result uncertainty 33.3%, probability of error 1%) in a concentration range from 0.5 – 50 µg L$^{-1}$. Due to deviations from linearity at low concentrations, different concentration ranges were used for diuron-desmethyl and terbuthylazin-2-hydroxy (5 - 100 and



25 - 1000 µg L$^{-1}$). Each calibration curve was determined as mean of three independent measurements. LOD and LOQ are given in Table 2. Table 3 gives an overview of analyzed substances.

**Table 2: LOD and LOQ of investigated substances with an enrichment factor of 4500 in surface water.**

| Substance | LOD [ng L$^{-1}$] | LOQ [ng L$^{-1}$] |
|---|---|---|
| Diuron | 0.22 | 0.78 |
| Terbutryn | 0.11 | 0.38 |
| OIT | 0.09 | 0.31 |
| Diuron-desmethyl | 1.33 | 4.67 |
| Terbuthylazin-2-hydroxy | 10.22 | 34.22 |
| Terbutryn-desethyl | 0.04 | 0.20 |
| Terbumeton | 0.04 | 0.13 |



**Table 3: Overview of analyzed substance. According to Hensen et al. (2018) and Paijens et al. (2019).**

| Substance | Molecular Formula | Chemical Structure | Log $K_{ow}$ at pH 7 | CAS-No. | PNEC [µg/L] |
|---|---|---|---|---|---|
| Diuron | $C_9H_{10}Cl_2N_2O$ | | 2.71 - 2.85 | 330-54-1 | 0.02 |
| Terbutryn | $C_{10}H_{19}N_5S$ | | 3.65 | 886-50-0 | 0.034 |
| Octylisothiazolinone (OIT) | $C_{11}H_{19}NOS$ | | 2.45 - 2.61 | 26530-20-1 | 0.013 |
| Diuron-desmethyl (diuron TP-219) | $C_8H_8Cl_2N_2O$ | | | 3567-62-2 | |
| Terbuthylazin-2-hydroxy (terbutryn TP-212) | $C_9H_{17}N_5O$ | | | 66753-07-9 | |
| Terbutryn-desethyl (terbutryn TP-214) | $C_8H_{15}N_5S$ | | | 30125-65-6 | |
| Terbumeton (terbutryn TP-226) | $C_9H_{19}N_5O$ | | | 33693-04-8 | |






## 2.5 Estimation of biocide emissions over two years

This section describes how total biocide emissions within a certain time period can efficiently be estimated based on a few chronological grab samples. In the study area this was possible for a roof facade that was repainted two years ago according to the house owners. Since it was situated on top of a flat roof, all storm water including biocide emissions was collected by

the rain downpipes. Since other sources were excluded by elution experiments, the net biocide emission BE (mg) from this single facade could be estimated multiplying average biocide concentrations C (mg $L^{-1}$) in the downpipes by the volume of roof runoff. Roof runoff was estimated multiplying flat roof area A (400 m²), by recorded rainfall P from the date of painting until time of sampling (1462 mm over 2 years). To account for evaporation losses, runoff coefficients RC from literature, i.e. 70 % for the roof terrace and 30 % for extensive roof greening (DIN 1986-100), were employed:

$BE = P * A * RC * C$                   (1)

BE was related to the initial amount applied on the facade BI (mg) to quantify biocide leaching percentage BL (%):

$BL = \left(\frac{BE}{BI}\right) * 100$                   (2)

BI was estimated multiplying the area of the newly painted facade AF (10 m²) by literature values of typical amounts of paint AP (0.2 L $m^{-2}$) including typical biocide concentrations CP (1500 mg $L^{-1}$ of diuron and terbutryn, 500 mg $L^{-1}$ of OIT, Sauer,

235   2017):

$BI = AF * AP * CP$                     (3)

## 3 Results and discussion

### 3.1 Standing water in swale (Step 1)

Data suggested that biocides emission was relevant in the investigated district, since measured concentrations of terbutryn and

diuron exceeded PNEC values (Fig. 5). Terbutryn and diuron were detected during all four events when standing water in the swale was sampled. Maximum concentrations in the first event (0.04 µg $L^{-1}$ terbutryn, 0.17 µg $L^{-1}$ diuron) exceeded PNEC values of surface water (0.034 µg $L^{-1}$ for terbutryn and 0.02 µg $L^{-1}$ for diuron). OIT was not detected in the swale. These concentrations were within the range found in other studies of urban storm water. Reported concentrations of terbutryn ranged between <10 and 360 ng $L^{-1}$ in storm water channels of a separated sewer system in the city of Berlin, Germany (Wicke et al.,

2015). There, diuron showed maxima up to 0.6 µg $L^{-1}$ and OIT up to 60 ng $L^{-1}$, while 2 µg $L^{-1}$ was found for diuron and OIT concentrations remained below4 ng $L^{-1}$ as mean concentration of storm water at the outlet of three catchments in Paris, Nantes and near Lyon (Gasperi et al., 2013). In another district of Freiburg, 2.8 km southwest, Hensen et al., (2018) found up to 5 ng $L^{-1}$ diuron, 160 ng $L^{-1}$ terbutryn, and up to 67 ng $L^{-1}$ OIT in a swale-trench system.

The TPs terbuthylazin-2-hydroxy and terbutryn-desethyl were detected in samples of the first two and the fourth event, diuron-

desmethyl of the second and third event. Terbumeton was detected only during the third event at very low concentrations of 0.25 ng $L^{-1}$. Diuron-desmethyl concentrations ranged between 0.2 and 4 ng $L^{-1}$, terbuthylazin-2-hydroxy between 8 and 48 ng





L$^{-1}$, and terbutryn-desethyl between 43 and 335 ng L$^{-1}$. Hensen et al. (2018) found concentrations of up to 23 ng L$^{-1}$ for terbuthylazin-2-hydroxy, 73 ng L$^{-1}$ for terbutryn-desethyl and 2 ng L$^{-1}$ for detect diuron-desmethyl in a swale-trench system located 2.8 km southwest. Differences in types and concentrations of detected substance between locations and events might

be due to different sources (e.g. newly painted facades), and different precipitation amounts and intensities that affect biocide emissions (Paijens et al., 2019). This may explain the highest biocide concentrations during the first rain event, which was the largest of the monitoring period (Fig. 4). Two of the four detected TPs (diuron-desmethyl and terbuthylazin-2-hydroxy) are described as most probably toxic or probably toxic (Hensen et al., 2020).

Swale water infiltrates to groundwater and contained pollutants can thus be an issue for groundwater quality (Burri et al.,

2019). Hensen et al., 2018 found diuron, terbutryn, OIT, diuron-desmethyl, terbuthylazin-2-hydroxy and terbutryn-desethyl in the shallow groundwater 2.8 km southwest. In our study, the groundwater table is significantly lower i.e. about 5 to 7 m below ground which might reduce the risk of contamination. Indeed, sporadic groundwater sampling in the vicinity of our swale did not show any detectable contamination. Still, biocides and their TPs remain a risk and there is a need for more intense monitoring of these substances in groundwater (Foster and Cogu, 2019).

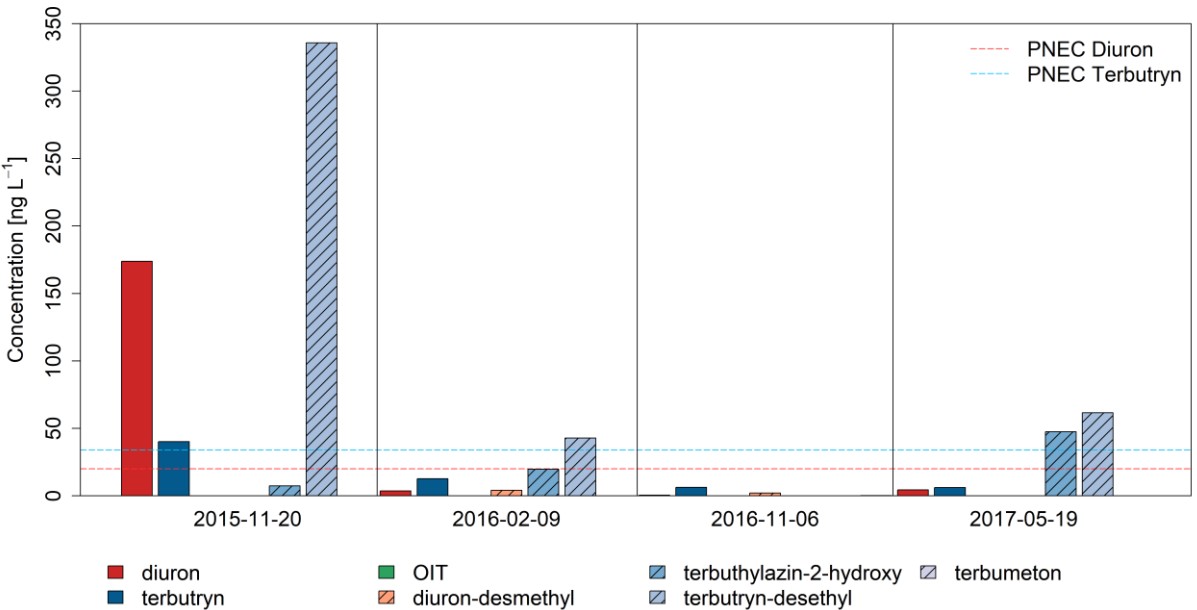


**Figure 5: Concentrations of monitored biocides of standing water in the swale during four events.**

**3.2 Source allocation (Step 2 - Part 1)**

**3.2.1 Facades**

Elution experiments at northwestern facades of houses 2, 4 and 5 showed different biocide composition of the wash off (Fig

6). Terbutryn was detected at one facade (F5), OIT at two (F2, F4), diuron not at all. Instead, diuron-desmethyl was detected at one facade (F2), which suggests in-situ photo transformation of the parent compound (Burkhardt et al., 2012). Terbuthylazin-



2-hydroxy was found at two (F2, F4) facades, terbutryn-desethyl at one (F5), terbumeton at none. Both detected TPs of terbutryn can be formed by photo transformation (Hensen et al., 2018). Detected TPs and biocides imply the application of terbutryn at all facades, diuron only at facade F2 and OIT at facades F2 and F4. Substances below LOD might have been part

of used renders or paints and may have been washed out or were not detected due to low concentrations. Although all houses were built at the same time, buildings show different color shades and sizes of protruding parts. Differences in detected TPs at the individual facades might be due to different color shades although its influence is reported to be rather low (Bollmann et al., 2018). Other factors influencing the differences in found TPs can include local wind conditions, UV exposure (Paijens et al., 2019) and, most probably, differences in application of paint and renders. Terbutryn was only found at F5 which pointed

to a different product used during construction also evident by different color and by very high concentrations of the TP terbuthylazin-2-hydroxy.

All houses were constructed thirteen years before the elution experiments. Studies under natural weather conditions found that most leaching takes place within the first months after painting. Thereafter, concentrations are reported to be lower and no longer as variable as before (Burkhardt et al., 2012; Bollmann et al., 2016). Although there is no experimental study

investigating facade leachate over a time period of more than a decade, it can be stated that leaching decreases significantly. Hensen et al., 2018 showed during a sprinkling experiment that leaching of biocides may occur even 15 years after initial painting. Thus, biocide leaching after 13 years should not be considered as surprise. Remarkably, diuron was found in swale samples but not in the facade elution experiments. Nevertheless, its use in paints and renders of the facades could indirectly be confirmed by detection of its TP diuron-desmethyl. Additionally, diuron might have been used at other facades in the area

which were not sampled here. Emitted OIT was very low at all facades and thus disappeared on its way to the swale. The elution experiments generally suggest that facades are a primary source of biocides and their TPs in this urban catchment and confirm the outcomes of the facade sprinkling experiments of Hensen et al. (2018).





**Figure 6: Photos of sampled facade sides and results of facade elution experiments at houses 2, 4 and 5.**

**3.2.2 Roofs and roof materials as source of biocides in storm water**

In all rain downpipes biocides were detected indicating the application of biocides on the flat roofs. However, concentrations of biocides and TPs largely differed between pipes and houses (Fig.7). Concentrations at house 4 (R4 - Pipe 1) exceeded those





at houses 1 and 2 by an order of magnitude. Due to these high concentrations, we sampled an additional pipe (R4 - Pipe 2) during following events. Both R4 pipes showed concentrations of a similar magnitude.

**Figure 7: Sampled events in rain water downpipes at three houses. Number after "R" refers to the number of the house.**

Concentration differences between houses can be attributed to different products used on different roofs, as previously discussed for the investigated facades (3.2.1). Additionally, houses differ in layout of their roofs (2.2) which might also lead to differences in biocide use and emission. Due to the limited number of investigated houses, general conclusions, if certain roof structures pose a higher risk for biocide emission than others cannot be drawn. Generally, biocides are used in roof



sealants, treated tiles, roof paints and bitumen roofing membranes (Paijens et al., 2019). Terbutryn and OIT may be included in paints for roof tiles (Jungnickel et al., 2008). Wicke et al. (2015) found different terbutryn concentrations in stormwater channels draining areas differentiated by their construction type and related these differences to the use of roof paints. In most studies, roof runoff is characterized by samples in stormwater channels of separated sewer systems (Burkhardt et al., 2011;

Wicke et al., 2015). If roof areas were investigated as a source for biocide emission, mostly experiments with bituminous roof sheets were carried out (Bucheli et al., 1998; Burkhardt et al., 2007; Wicke et al., 2015). To the best of our knowledge we are not aware of a study that detected terbutryn, diuron or OIT in rain water downpipes. In our study, diuron and diuron-desmethyl were found in high concentrations (1 µg L$^{-1}$ and 320 ng L$^{-1}$ respectively) in the two rain downpipes of house 4. These findings indicated the presence of an important source that had to be identified by leaching and elution experiments (Fig. 8).

Leaching tests of the wooden roof terrace taken from house 4 showed no biocides or TPs present in the extraction solution. Elution experiments of various roof materials showed very low concentrations of terbutryn (<1 ng L$^{-1}$) (Fig. 8a), while OIT was found in the railing, in the roof foil and in the roof access (max. 12 ng L$^{-1}$). These low concentrations did not suggest a primary source as it was indicated by the findings in the rain downpipes. However, elution experiments at parts of the inner roof facade yielded very high concentrations (2.7 µg L$^{-1}$ diuron, 2.6 µg L$^{-1}$ diuron-desmethyl and 1.9 µg L$^{-1}$ OIT, Fig. 8b). This

inner roof facade exists at all houses, but at house 4 a 5 m² westward facing part (approximately 10 m long and 0.5 m high) was repainted in August 2018. Its western exposure suggests a higher emission rate of biocides due a higher amount of wind driven rain at the weather side (Vega-Garcia et al., 2020). Diuron-desmethyl was presumably formed as a photo TP at this facade. Additionally, diuron, diuron-desmethyl and OIT were found on the northern side of the roof facade, at an area of about 10 m² (Fig. 8c). Concentrations were much lower (diuron: 48 ng L$^{-1}$, diuron-desmethyl: 29 ng L$^{-1}$, OIT: 25 ng L$^{-1}$) than at the

newly painted area. On the southern side of the roof facade, only OIT was detected in low concentrations (7 ng L$^{-1}$). Terbutryn was only found in two elution experiments of the roof material and roof facade. However, terbutryn and two TPs were found at all investigated rain downpipes in low concentrations. It may thus be assumed that terbutryn is used in the railing and possibly also in other roof materials and is still leached. With the current test design this could not be determined in more detail.



**Figure 8: Schematic view of roof area of House 4 with sampling spots. Results of sampling for (a) roof materials include roof balustrade, railings, roofing foil, roof access, roof cladding, elevator shaft foil and grass foil, (b) newly painted roof facade and (c) the old roof facade. Lines show minimum and maximum of sample category.**

### 3.2.3 Estimation of biocide emissions over 2 years

This section evaluates the role of newly painted facades for biocide emission from house 4. Findings at the newly painted roof facade at house 4 suggested a locally limited source with high impact. Other facades could be disregarded, since elution tests showed biocide emissions an order of magnitude lower than for the newly painted roof facade (Figs. 6, 8c). This conforms with existing knowledge, since most leaching takes place within the first months after painting (Burkhardt et al., 2012). Table 4 shows the results of calculated two-year biocide emission (BE) for diuron, terbutryn, OIT and diuron-desmethyl. Biocide leaching percentage for diuron-desmethyl was related to its parent compound diuron. Although we did not account for factors





influencing leaching (e.g. variable meteorological conditions, unknown composition of the used paint, leaching kinetics and other parts of the roof that might contribute to leaching), our estimated leaching percentages (BL) compared to reported values from experiments under natural weather conditions. Burkhardt et al., (2012) showed a leaching of 3.5 % for terbutryn, 13.4 % of diuron and 3.9 % of OIT, 12 months after initial painting. This matches our results, since diuron also showed highest
leaching percentage. Other studies quantified leaching 18 months after initial painting and reported 3 % for terbutryn (Bollmann et al., 2016) and 2.8 % for OIT (Bollmann et al., 2017b). The existing studies investigated artificial facades at artificial walls (Bollmann et al., 2016; 2017b) or facades at a model house (Burkhardt et al., 2012) but no real-world case. Our study thus confirms the realism of the artificial experiments.

We have to admit that the accuracy of these estimations is limited due to various reasons. First, we only applied literature
values to quantify the amount of used color and its biocidal content. Second, we used precipitation data from a weather station located 5 km away. Third, we did not have information about wind driven rain at the investigated facade although this is an important factor regarding biocide emission (Burkhardt et al., 2012). Forth, we disregarded variable light intensity or dry periods between rain events. Fifth, we approximated the receding biocide concentrations in rain downpipes (Fig. 7) by mean values of four events. Sixth, we did not have information about initial biocide leaching prior to our sampling period and
extrapolated mean concentrations to the entire period of two years. Regardless of these uncertainties we arrived at realistic values why we consider our approach promising for an initial estimation of relevant biocide sources by a limited number of samples.

**Table 4: Estimated 2-year biocide emission from 10 m² rooftop facade at house 4 with BE as net biocide emissions, BI initial biocide concentration at facade and BL biocide leaching percentage.**

| Biocide | Equation (1) BE [mg] | Equation (3) BI [mg] | Equation (2) BL [%] |
|---|---|---|---|
| Diuron | 155 | 3000 | 5.2 |
| Diuron-desmethyl | 68 | 3000 (diuron) | 2.3 |
| Terbutryn | 17 | 3000 | 0.6 |
| OIT | 12 | 1000 | 1.2 |

### 3.2.3 Entry pathways of biocides from buildings into the swale (Step 2 - Part 2)

The three investigated drainage pipes S9 (surface water), P10 and P11 (percolation water) are entry pathways for emitted biocides into the swale. Terbutryn was detected in all pipes during all four sampled events, while diuron was found in percolation water pipe P11 during three and in surface water pipe S9 during one event (Fig. 9). As in the swale (Fig. 5), OIT was not detected at all, probably due to fast degradation in the soil passage (Bollmann et al., 2017b).





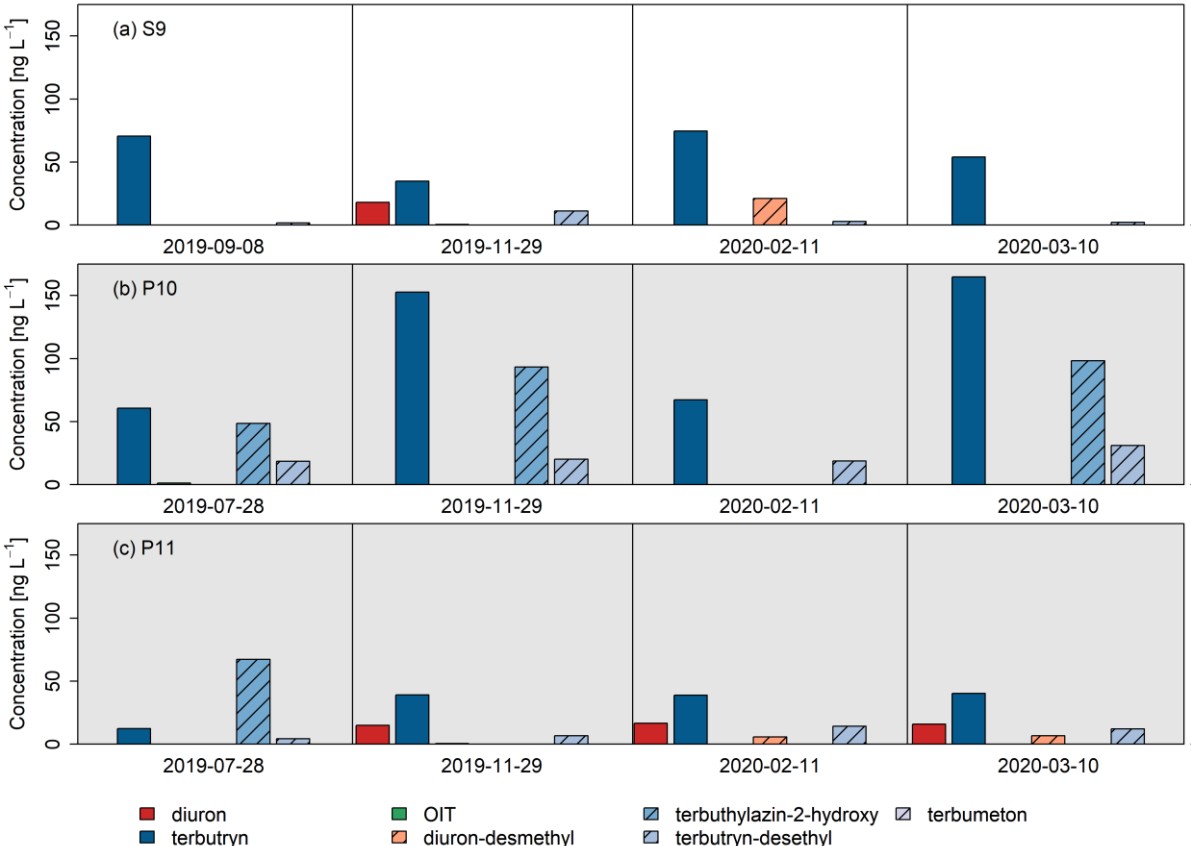


**Figure 9: Samples in drainage pipes during four events. (a) Pipe S9 collects street runoff. (b) Pipes P10 and (c) P11 collect infiltrated water around the houses (percolation water).**

Detected biocides in percolation water pipes P10 and P11 were different probably due to different paints and renders used in the corresponding connected areas. The biocide concentrations of individual products may also vary (Sauer, 2017). Another

reason might be a different dilution before reaching the outlet of the drainage pipe due to possible differences in the pipe system leading to the individual drainage pipe outlets. Additional sources may also affect the system, since infiltrated water from gardens and terraces around the building accumulates on top of the underground parking garage and reaches the drainage pipes. This could include biocides used on terraces or garden furniture, although inhabitants denied having used biocides. Diuron and terbutryn are authorized for use in coatings, which includes terraces or furniture coating (European Parliament and

Council, 2013).

Diuron-desmethyl was detected only during one event in surface water pipe S9 and during two events in percolation water pipe P11. Concentrations of diuron-desmethyl were lower in P11 than in S9. This might be explained by higher dilution effects in the percolation water pipe P11 or by enhanced degradation during soil passage. Diuron-desmethyl is reported to be a photo TP that possibly can be formed by microorganisms in soil additionally (Hensen et al., 2018.).





TPs of terbutryn were detected in all pipes, although there was a difference between surface water and percolation water pipes regarding the number of detected terbutryn TPs. We detected terbutryn-desethyl in all three drainages during all events. The different appearance of the other two TPs of terbutryn implied different degradation processes in pipes S and P. Terbuthylazin-2-hydroxy was only found in percolation water pipes P10 and P11 (with soil passage) and not in S9 (without soil passage). This suggests preferred formation of terbuthylazin-2-hydroxy by biodegradation, although photo transformation could not be

excluded, since both terbutryn TPs were already found at the sources (Fig. 5, 6). Concentrations in the swale were in a similar range, which suggests similar input concentrations and degradation processes. Terbumeton was not found in samples in the drainage pipes, though detected in the swale.

For terbutryn, concentrations of both the parent compound and its TPs were highest in percolation water pipe P10. Biochemical transformation in soil is important and TPs are obviously formed, our findings suggest that also the parent compound poses a

risk to groundwater by diffuse infiltration of contaminated runoff from urban settings. Much less transformation of terbutryn takes place along surface pathways, represented by surface water pipe S9 where overall concentrations of TPs were smaller and terbuthylazin-2-hydroxy was not found.

### 3.4 Use of two-step approach for efficient monitoring

Some studies used extensive flow proportional sampling to calculate loads (Bollmann et al., 2014b; Wicke et al., 2015; Paijens

et al., 2021). To limit analytical costs, we did not assess overall biocide loads but rather stated with single samples at the outlet to document the overall relevance of biocide emission and then focused on sources and transformation along pathways. Here particularly TPs can give additional information as shown in Fig. 10, which gives a qualitative overview of detected substances at the different sampling points. Future studies might focus on loads of biocides and TPs in order to understand the cumulated entry into the environment over longer time periods. Sampling methods such as passive sampler may also help to reduce the

expenses and monitor more substances over a larger area (Gallé et al., 2020) or multiple catchments (Pinasseau et al., 2020).



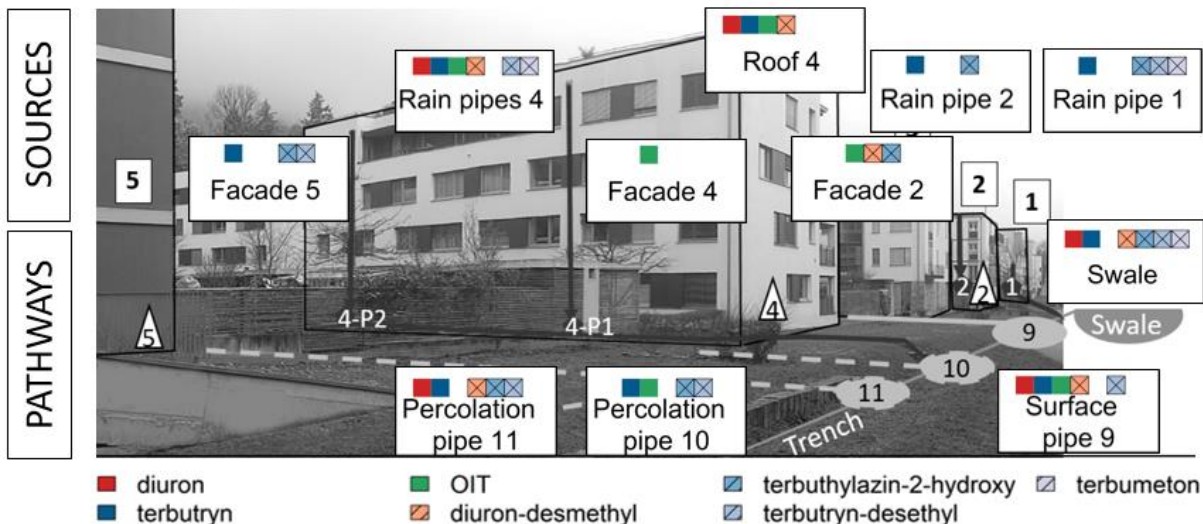

**Figure 10: Qualitative overview of biocides and their TPs in the investigated water infrastructures. Shown colors at sampling points represent detected substances. For legend of background picture, refer to Fig. 3.**

We also showed which infrastructure or area is important and should be considered for reducing the risk of biocide pollution. We identified sources both by sampling rainwater downpipes and by conducting artificial elution experiments. This might be an advantage when considering the transfer of our approach to other catchments due to an easy adaptation to existing buildings. We found additional biocide sources besides the main building facades by targeted elution experiments of various roof materials on a contaminated flat roof. This stresses the fact to consider the entire building including roof areas as a potential

source of biocides. In our case, a small re-painted roof facade was identified as a major biocide source, even 13 years after construction of the building had been completed. Therefore, existing buildings must be regarded as continuous biocide sources, not only originating from old facades as shown by Hensen et al., 2018, but also due to sporadic repair on restricted areas. The location of the roof facade on top of a flat roof permitted the estimation of long-term biocide leaching with only a minimum number of samples at the rain downpipes. The obtained results agreed with previous studies of artificial walls under natural

weather conditions (Burkhardt et al., 2012; Bollmann et al., 2016; Bollmann et al., 2017b). Hence, we advocate the use of existing urban building infrastructure, i.e. flat roofs and rainwater downpipes, to efficiently collect long-term realistic data on biocide wash-off with only a minimum sampling effort. This approach cannot be used to close mass balances, since there is not enough information on initial inputs or complete biocide loads during individual events. With more data available, such as wind-driven-rain or initial biocide usage, such data could also be used to calibrate existing physically based models on potential

leaching from buildings (Tietje et al., 2018) or from larger urban areas (Coutu et al., 2012).

Finally, we investigated entry pathways and results suggest effects of the soil passage on biocide breakthrough to groundwater and on biocide transformation. Again, we used existing urban infrastructure, in this case the collection of areal infiltration by a drainage system on top of an underground parking garage. Through this our study adds to the few available studies that investigate the risk of biocide entry into groundwater in urban areas (Hensen et al., 2018) .





Our approach can principally be adapted to other areas of commonly built modern urban districts, where separated sewer systems, thermal insulation of buildings and modern architecture with flat roofs and limited roof overhang prevail and promote biocide emission. With a limited number of samples and analyzed substances especially small-scale districts can be characterized regarding their potential risk of biocide emissions. This shows new possibilities to reduce the pollutant entry in a targeted manner and to take measures at the source (Kümmerer et al., 2018).

**4 Conclusion**

Following the introduced method, we first confirmed the relevance of biocide emission in the investigated urban system by spot sampling during four events in a swale. Receiving urban infiltration system (e.g. swales, swale-trench systems, retention ponds) at the outlet of a catchment generally provide an integrated signal of the aquatic system of a larger area. Thereafter, source areas were identified, again with a limited number of samples. Artificial elution experiments confirmed expected
sources, i.e. facades. Some facades showed only TPs but no biocides which were presumably washed off before sampling. Therefore, TPs may help to identify previously used substances and can thus complete the picture of biocide use on facades. Besides facades, we found additional sources through sampling of rain downpipes from flat roofs. In our case, high concentrations in one downpipe helped to identify a small recently painted roof facade as a primary biocide source. We therefore advocate the sampling of rain downpipes as this can help to identify additional sources and also facilitate estimations
of emitted biocide loads, since volumes of roof runoff can easily be estimated when rainfall is known. Monitored drainage pipes characterized entry pathways from buildings to the swale and suggested differences in biocide transformation due to soil passage. Yet, all pipes showed concentrations of terbutryn, regardless of a pathway through soil or not. This shows the risk that biocides are not necessarily degraded on their way to groundwater. For surface water our study directly confirmed the environmental risk of biocide use, since concentrations at the outlet of our urban catchment exceeded PNEC values. This
means that biocides were emitted into the urban environment more than a decade after construction had ended. Hence biocide pollution is not limited to newly built areas but a continuous and omnipresent problem relevant for all urban areas. We think that our parsimonious approach can easily be adopted to other cities to evaluate the risk of biocide pollution.



**Financial support:** This research was funded by the EU within the European Regional Development Fund (ERDF), support
measure INTERREG V in the Upper Rhine in the project 5.3 NAVEBGO (Sustainable reduction of biocide inputs to
groundwater in the Upper Rhine region).

**Competing interests:** The authors declare that they have no conflict of interest.

**Author contributions:** FL sampled and visualized the measured data and prepared most of the manuscript in cooperation with
JL and OO. JL, FP, FL and MB were responsible for the conceptualisation. LS, OO and KK facilitated the sample analysis
and LS wrote the section on analytical methodology. All authors were involved in the editing and revision of the manuscript.





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
