# Peer review of "Sources and pathways of biocides and their transformation products in urban stormwater infrastructure of a 2 ha urban district"

_Hydrology and Earth System Sciences, 2021_

## Referee Comment (RC1)

**General comments**

The authors describe a strategy to identify sources of biocides and transformation products in an urban area. Analysis of transformation products in addition to original biocides completed information on environmental impact of constructions, but also indicated former use of substances that were not detected in the samples.

The described study expands knowledge on the distribution of biocides and transformation products in urban areas, which is urgently required in order to develop strategies to minimize emissions into the environment.

Related work is comprehensively considered, and own contributions are clearly indicated. Title and abstract reflect the content of the paper. The overall presentation is well structured, clear and descriptively illustrated by figures on the investigation area. The conclusions are supported by the obtained results.

The paper is recommended for publication in HESS with minor revisions.

**Specific comments**

- Line 48 Bollman et al. 2016 and 2017 b report on transformation of terbutryn and OIT on facades, please add these references here
- Line 50 231 days: this value was a result of Bollmann et al. 2017a
- Line 57 There are numerous papers that report on transformation products of diuron, terbutryn and OIT. This statement could be related to façade coatings to avoid a long list of references. However, the papers of Bollmann et al. 2016 and 2017b should be cited here at least. It may also be important that modern investigations on degradation products of diuron is usually limited to diuron-desmethyl although there are reports on a number of other transformation products (e.g. Jirkovský et al.: Photolysis of Diuron. Pesticide Science 50/1 (1997) 42-52 and other reports, see also Hensen et al. 2020). Possibly, other degradation products of diuron have been overlooked (not only in this study).
- Line 69 Biocides and their TPs can enter the environment only in case of driving rain to the surface (not generally during rain events).
- Line 70 Please explain how the elution experiments were performed on roof materials from house 4 especially in case of horizontal orientation.
- Line 196 Please add information on the recovery of the SPE procedure for the analytes.
- Line 230 Estimation of net BE: The estimation of net biocide emissions according the given formula cannot be correct. Several reasons why this is incorrect are discussed later in the text (Line 349). Please change wording under 2.5 to clarify that this calculation is a rough estimate with certain reservations.
- Line 255 Differences in substance patterns are probably also caused by different intensity of UV radiation.
- Line 277 Different patterns of transformation products depending on different pigments were observed by Urbanczyk et al. 2019 (Influence of pigments on phototransformation of biocides in paints. Journal of Hazardous Materials 364 (2019) 125-133).
- Line 435 Missing biocides in the samples is not necessarily explained by former wash-off. Water solubility of most transformation products is probably higher than water solubility of the biocides. Therefore, the TPs should be washed off easier than the biocides. Probably, biocides that were available on the surface were almost completely transformed. It cannot be excluded, that biocides are still present in deeper layers of the materials that were not reached during the very short elution experiment.

Line 444 For environmental risk assessments it is urgently required whether PNEC values are occasionally or permanently exceeded. The data for the swale indicate that the PNEC values for diuron and terbutryn were exceeded in one out of four samples from the swale. Please clarify this statement.

**Technical corrections**

- Title please add a blank between '2' and 'ha'
- Line 14 use capital letters for Central and Northern Europe (also in the following text)
- Line 64 please delete either 'and' or 'but'
- Line 95 Please check the number of samples (52). The number of samples described in Table 1 amounts to 49. 3 samples from artificial experiments on facades and 20 samples from artificial experiments on roof materials from house 4 and x samples from a leaching test on the wooden terrace are mentioned in the text.

Possibly, the origin of the samples can be mentioned here (collected in the swale, rain downpipes and drainage pipe; from elution experiments on facades and roof materials from house 4 and a leaching test on the wooden terrace).

- Line 97 please correct: 'selected'
- Figure 1 please correct: Step 2 Part 1 Identify source areas (instead of 'sources')

Method: the phrase 'elution experiments at selected infrastructures' would facilitate understanding the different methods mentioned here

- Table 1 The information '(Duplicates >1)' seems to be unnecessary and rather confusing.
- Line 246 please add a blank between 'below' and '4'
- Line 250 please add 'in samples' after 'desmethyl'
- Line 253 please delete 'detect'
- Line 432 please correct: 'systems'
- Line 500 please add a link
- Line 510 please add a link

---

## Referee Comment (RC3)

**Comments on HESS-2021-143**

**General comments**

This article deals with the sources and pathways of some biocides and their transformation products from building runoff in a 2 ha urban district. The manuscript is overall of good quality, well written and presenting interesting data with original sampling campaigns to study the biocide emissions from a small urban districts.

My main reservation is with the section "Estimation of biocide emissions over two years". The description of the method (paragraph 2.5) and the conclusion on this subject must be qualified. The method gave a rough estimate (even if interesting) but the comparison to the literature is only informative and does not bring proof of the accuracy of the evaluation, the initial stock of biocides not being necessarily the same. The presentation of the results of this part is however honest (paragraph 3.2.3).

The paper is recommended for publication with some minor revisions.

**Specific comments**

Title: Perhaps specify in the title that it deals with stormwater

Abstract: The last sentence of the abstract present obvious conclusions and not informative. It seems logical that by sampling in a targeted way, a better identification of the sources is obtained. Perhaps you can refine this conclusion and add some more concrete and precise results.

Line 57: Please explain the choice of the TPs, why just these 3 compounds?

Paragraph 2.1: the tow-step approach is well presented and convincing, but the long period between the first campaign (step 1 in 2015-2017) et the last one (step 2 in 2019-2020) raises the question of the comparability of the campaigns between them. Why did you not sample the swale system during the second campaign in 2019-2020 to verify the stability of the concentrations in the swale? Justify this point.

Line 117: you said that the last paint was in 2007 and after it is indicated that a façade was painted in 2018 (line 223). It is unclear.

Line 121: Is there always water in the swale or is it dry during dry weather?

Line 153: how are sampled the roof, façade and pipe samples? Are they representative of the entire rain events? What about the first flush? You should add details about the sampling and its representativeness.

Line 153: how the water is sampled? You said during the sample, why not at the end of the sample?

Line 163-165: did you analyse the representativeness of the sampled events in relation to the classical pluviometry?

Figure 4: I am not sure that this figure is really informative since we are not able to read the rainfall for each sampled event. Perhaps put it in supplementary materials

Line 176: why did you not test solar panel elution? Do you think that they could emit biocides?

Line 177-178: n=1 seems insufficient to conclude.

Chemical analysis: I would recommend to present the analytical validations (as extraction recoveries) and the analytical uncertainties to validate the SPE extractions and the quantification.

Paragraph 2.5: I am not really convinced by the methodology presented by this paragraph because the concentration used does not take into account the temporal evolution of emitted concentrations over time due to ageing or depletion of the stock in the material, or does not present an argument from the literature to overcome this. What verification have you implemented to justify the word "efficiently" in line 222. Justify the use of an average biocide concentrations to calculate BE. Moreover, why the used samples were not described in the 2.3 section? What is the number of the samples and the representativity? What is the sampling frequency? What is the variability of the measured concentrations? Does the concentration vary in time? Decrease? To prevent the reader's doubts, a part of the explanation from line 349 to 355 could be used in the methodology presentation and the fact that the estimated BE will be compared to the literature.

Paragraph 3.1: I am wondering if 4 sampled events are sufficient to assess the variability of the concentrations in the swale, especially since only one PNEC exceedance is observed to justify the continuation of the study. Why don't you continue to sample the swale in the second part of the study?

Line 298: You said that you sampled an additional pipe (R4-2) because R4-1 exceeded R1 and R2 by an order of magnitude but you have no result for R1 and R2 before the first sampling of R4? I don't understand.

Figure 8: Precise if it is mean or median values in the legend

Line 320 and following: It is not clear if the difference of concentrations is due to the new paint or to the exposition.

Line 363/364: you explain that OIT was not detected due to its degradation in soil but for S9 (surface water pipe), water is not percolated through the soil? How do you explain to not found OIT in S9 samples?

Line 413: you have to qualify this sentence because your method gave a rough and short term estimate (even if interesting). the comparison to the literature is only informative and does not bring proof of the accuracy of the evaluation, the initial stock of biocides not being necessarily the same.

**Technical corrections**

Figure 1: Step 2: perhaps precise "Elution and leaching test experiments"

Line 97: two -s at "sselected"

Line 114: The capital letter at "Area" is unnecessary

Line 118: perhaps add a -s at "diverse use"?

Line 134: I find that "surface water pipe" do not describe well the type of water sampled. It looks like surface water that has been sampled. Perhaps "surface runoff pipe" would be more meaningful

Figure 5: you could cut the ordinate-axis to better present the lowest concentrations

Line 246: space is missing between below and 4

Line 245, 246 and 247: the sentence is not simple to understand

Line 254: substanceS

Line 296: perhaps add a coma after "in all rain downpipes"

Figure 7 (d): perhaps precise "non sampled event". The use of one single scale is understandable but does not allow to read the concentrations for R1 and R2

Line 321: due "to"?

Line 579: "TEXTE"?

---

## Author Comment (AC1)

We would like to thank Ute Schoknecht for the time she has taken to read our manuscript and her helpful comments to improve it. In the following section we are going to repeat the points brought up (in grey italic letters) and subsequently respond to them:

*Specific comments*

Line 48 — *Bollmann et al., 2016 and Bollmann et al., 2017b report on transformation of terbutryn and OIT on facades, please add these references here* – will be added

Line 50 — *231 days: this value was a result of Bollmann et al., 2017a-* will be corrected

Line 57 — *There are numerous papers that report on transformation products of diuron, terbutryn and OIT. This statement could be related to façade coatings to avoid a long list of references. However, the papers of Bollmann et al., 2016and Bollmann et al., 2017b should be cited here at least. It may also be important that modern investigations on degradation products of diuron is usually limited to diuron-desmethyl although there are reports on a number of other transformation products (e.g. Jirkovský et al.: Photolysis of Diuron. Pesticide Science 50/1 (1997) 42-52 and other reports, see also Hensen et al. 2020). Possibly, other degradation products of diuron have been overlooked (not only in this study).*

We will revise this paragraph so that its focus on selected TPs and on façade coatings becomes clearer. Regarding degradation products Diuron, we will include different relevant TPs. Thank you for your additional literature suggestions. We were not aware of the study by Jirkovský et al., 1997 and will add this to the manuscript.

Line 69 — *Biocides and their TPs can enter the environment only in case of driving rain to the surface (not generally during rain events). -* We will clarify our statement.

Line 70 — *Please explain how the elution experiments were performed on roof materials from house 4 – especially in case of horizontal orientation.*
This comment refers to line 170. We will include more details on the elution experiments as follows: Most roof materials (i.e. Roofing foils, roof access, roof cladding, elevator shaft foil and grass foils) were tested where their orientation was vertical, e.g. around vertical orientated pipes or shafts. We conducted experiments like on facades. Railings were accessible from all sides so we conducted elution experiments by setting a container underneath them. At the railings there were certain limitations regarding area poured with water so this might not be comparable to experiments at the facades. We dismantled parts of the wooden terraces so we were able to access the substructure and set a container underneath the wooden bars. This way we could perform leaching experiments on a horizontal surface the same way as on the facades.

Line 196 — *Please add information on the recovery of the SPE procedure for the analytes.*
Recovery was determined by spiking water samples with 1 mg $L^{-1}$ of analytical standard and was found to be 97.7 % (Diuron), 88.5 % (Terbutryn) and 93.5 % (OIT), 85.0 % (Diuron-desmethyl), 66.2 % (Terbumeton ), 50 % (Terbuthylazin-2-hydroxy) and 92 % (Terbutryn-desethyl) (Hensen et al., 2018).

Line 230 — *Estimation of net BE: The estimation of net biocide emissions according the given formula cannot be correct. Several reasons why this is incorrect are discussed later in the text (Line 349). Please change wording under 2.5 to clarify that this calculation is a rough estimate with certain reservations.*
We are aware that our estimation has many limitations and only gives a very rough estimate. We will clarify this also here.

*Line 255*    *Differences in substance patterns are probably also caused by different intensity of UV radiation.*

We will add this as a possible explanation also here. This nicely fits into the paper, since we have also mentioned the impact of UV radiation in the introduction and in section 3.2.1.

*Line 277*    *Different patterns of transformation products depending on different pigments were observed by Urbanczyk et al. 2019 (Influence of pigments on phototransformation of biocides in paints. Journal of Hazardous Materials 364 (2019) 125-133).*

Thank you for your remark and reference. We will add this fact as an additional explanation.

*Line 435*    *Missing biocides in the samples is not necessarily explained by former wash-off. Water solubility of most transformation products is probably higher than water solubility of the biocides. Therefore, the TPs should be washed off easier than the biocides. Probably, biocides that were available on the surface were almost completely transformed. It cannot be excluded, that biocides are still present in deeper layers of the materials that were not reached during the very short elution experiment.*

Thank you for clarifying. We will add this explanation to the discussion on façade (Section 3.2.1). We will also shorten the original sentence here.

*Line 444*    *For environmental risk assessments it is urgently required whether PNEC values are occasionally or permanently exceeded. The data for the swale indicate that the PNEC values for diuron and terbutryn were exceeded in one out of four samples from the swale. Please clarify this statement.*

Thank you for pointing this out. We did not aim for a complete environmental risk assessment and will clarify our statement. We are aware that a limited number of measurements cannot give information about long-term environmental risk. However, we will stress the fact that biocide pollution remains an issue after more than a decade after construction has ended.

*Technical Corrections*

*Title*    *please add a blank between '2' and 'ha'* - will be corrected

*Line 14*    *use capital letters for Central and Northern Europe (also in the following text)* - will be corrected

*Line 64*    *please delete either 'and' or 'but'* - *will be corrected*

*Line 95*    *Please check the number of samples (52). The number of samples described in Table 1 amounts to 49. 3 samples from artificial experiments on facades and 20 samples from artificial experiments on roof materials from house 4 and x samples from a leaching test on the wooden terrace are mentioned in the text. -*
*Possibly, the origin of the samples can be mentioned here (collected in the swale, rain downpipes and drainage pipe; from elution experiments on facades and roof materials from house 4 and a leaching test on the wooden terrace).*

We will update the table and make it clear.

*Line 97*    *please correct: 'selected'* - will be corrected

*Figure 1*        *please correct: Step 2 – Part 1 Identify source areas (instead of 'sources')* – will be corrected

        *Method: the phrase 'elution experiments at selected infrastructures' would facilitate understanding the different methods mentioned here* – will be corrected

*Table 1 The information '(Duplicates >1)' seems to be unnecessary and rather confusing.*
Thank you for your suggestion. Indeed, this might be a bit confusing. We will change it to 'Samples taken per event (0 = no samples, 1 = 1 sample, 2= duplicate samples taken during one event)'.

*Line 246*        *please add a blank between 'below' and '4'* - will be corrected

*Line 250*        *please add 'in samples' after 'desmethyl'* - will be corrected

*Line 253*        *please delete 'detect'* - will be deleted

*Line 432*        *please correct: 'systems'* - will be corrected

*Line 500*        *please add a link*
This refers to line 501 where a link is missing. We will add the link accordingly (https://echa.europa.eu/de/information-on-chemicals/biocidal-active-substances).

*Line 510*        *please add a link* – will be added (https://www.thesourcemagazine.org/urban-groundwater-mobilising-stakeholders-to-improve-monitoring/ )

**References**

Bollmann, U. E., Fernández-Calviño, D., Brandt, K. K., Storgaard, M. S., Sanderson, H., and Bester, K.: Biocide Runoff from Building Facades: Degradation Kinetics in Soil, Environmental science & technology, 51, 3694–3702, doi:10.1021/acs.est.6b05512, 2017a.

Bollmann, U. E., Minelgaite, G., Schlüsener, M., Ternes, T., Vollertsen, J., and Bester, K.: Leaching of Terbutryn and Its Photodegradation Products from Artificial Walls under Natural Weather Conditions, Environmental science & technology, 50, 4289–4295, doi:10.1021/acs.est.5b05825, 2016.

Bollmann, U. E., Minelgaite, G., Schlüsener, M., Ternes, T. A., Vollertsen, J., and Bester, K.: Photodegradation of octylisothiazolinone and semi-field emissions from facade coatings, Scientific reports, 7, 41501, doi:10.1038/srep41501, 2017b.

Giacomazzi, S. and Cochet, N.: Environmental impact of diuron transformation: a review, Chemosphere, 56, 1021–1032, doi:10.1016/j.chemosphere.2004.04.061, 2004.

Hensen, B., Lange, J., Jackisch, N., Zieger, F., Olsson, O., and Kümmerer, K.: Entry of biocides and their transformation products into groundwater via urban stormwater infiltration systems, Water research, 144, 413–423, doi:10.1016/j.watres.2018.07.046, 2018.

Jirkovský, J., Faure, V., and Boule, P.: Photolysis of Diuron, Pestic. Sci., 50, 42–52, doi:10.1002/(SICI)1096-9063(199705)50:1<42:AID-PS557>3.0.CO;2-W, 1997.

---

## Author Comment (AC2)

We would like to thank the anonymous referee for the time he/she has taken to read our manuscript and his/her helpful comments to improve it and gain clarity. In the following section, we are going to repeat the points brought up (in grey italic letters) and subsequently respond to them:

*Abstract*
*Line 18-20:    This sentence is very difficult to read please divide it into two sentences.* – We will change this sentence accordingly.

*Line 23-24:    Revise the sentence sintaxis, very difficult to read-*– We will change this sentence accordingly.

*Line 25:    Delete "for" after "allows".*– will be corrected

*Introduction*

*Line 40-44:    Please divide this sentences into two or serveral sentences, is to difficult to read.* We will revise these sentences to make them clearer.

*Line 46-47:    This information is already included in Table 3, please delete.* – will be corrected

*Line 50:    Delete "for example"* - We will change this to "among others" as we mention the most common influences.

*Line 57:    Delete the comma, instead place TPs in parenthesis.* – will be corrected

*Line 57-58:    This sentence should be at the end of the paragraph.* – We will revise this paragraph.

*Line 77:    Geometry?* – *We will add this and refer to Burkhardt et al., 2012.*

*Line 81:    "Studies have confirmed…"*– will be corrected

*Line 86:    Please delete the comma and better add swale-trench system in parenthesis.* – We will correct the comma. We are aware that various terms exist for urban storm water infiltration systems. We will define the term swale-trench system more clearly by referring to the study of Hensen et al., 2018.

*Line 88-89:    This sentence is very difficult to read.* – We will improve this sentence.

*Methods*

*Line 109-110:    MEC/PNEC where chosen for what? Which criterion? Relevance threshold?*
In a first step, the objective was to determine the relevance of biocides in our study area. We chose MEC/PNEC >1 as a common threshold for environmental risk assessment. We are aware that our study is not a complete environmental risk assessment. Our objective rather was to have a defined starting point for further investigations. See also our answers to referee #1 Ute Schoknecht.

*Line 115: Please add coordinates* - We will add coordinates: 47° 59N   7° 51E.

*Line 126-128:  Please add here the total facade area if possible, or size of the buildings and roof top total area approx. Its important to have an idea of the biocide loads from each of the buildings or from the total building complex.*
Thank you for your suggestion. We will add information on the geometry of the buildings.

*Line 131-134: Please add the pipeline/drainage material*- We will add the material.

*Line 152-155: Please add the total amount of samples within the test period*
We will change Table 1 so that the number of samples becomes clearer. See also our answers to referee #1 Ute Schoknecht.

*Line 180: Is this leaching test out of norm/standard (i.e DSLT) or it is a self fabricated test? If it is, please argue why you do the leaching test that way.*
*The aim here was to account if any leaching takes place at all. That is why we used a self-fabricated test. The first leaching test did not show any biocide concentrations. Further elution experiments at other parts of the wooden terrace confirmed that this was not a biocide source. We will discuss this in the updated manuscript.*

*Line 195: Instead of "measurement" use "analysis".*— will be corrected

*Table 3: Please add water solubility, half-life time, molecular mass and lethal dose.* — We will add this information.

Results
*Line 245: "There, diuron showed maximum concentrations of…"*— will be corrected

*Line 255: Please add weather data elsewhere in studied area/sampling site (methods section). Here you argue about weather conditons in the area but there is no information of it prior this argumentation.*
We did not have a weather station in the immediate district, but relied on a weather station about 5km away from the study area, see 2.3.1. We will stress this in the updated methodology. In Fig. 4 used rainfall data to illustrate rainfall magnitudes during the sampled events. We will add the amount of precipitation in an updated figure.

*Section 3.2.1: Does the impinged water volumes have an influence in the leaching concentrations? All the facades received the same amount of water? Are collected runoffs in the same order? It is important to mention this since the leaching amount of substances is also dependant on the contact water volume. Higher the runoff volume, higher the substance load.*

*Please mention in this section something about the contact water volume, it is an important parameter into consideration when talking about substance leaching of facades. Consider biocide loads (mg/m² or µg/m²) in this section, since this measurement is important for environmental evaluation properties of any construction site.*
Thank you for your comment. We are aware that the impinged water volume has an influence on the leaching concentrations. We conducted the elution experiments as similar as possible to reduce such influences. Collected runoff volumes were in the same order of magnitude, about 1L.
We sprinkled about 1L across 0.25m² and collected the entire runoff (see 2.3.2). We repeated these experiments twice at each investigated façade and found similar concentrations in the obtained duplicates. We will clarify this point in the discussion of the updated manuscript and stress that the results should not be evaluated quantitatively but rather qualitatively in a sense that a specific biocide was detected or not. This also due to the fact that information on initial biocide loads could not be determined for all buildings (see 2.2).

*Line 421:        Please delete "Again"`*— will be corrected

**References**

Burkhardt, M., Zuleeg, S., Vonbank, R., Bester, K., Carmeliet, J., Boller, M., and Wangler, T.: Leaching of biocides from façades under natural weather conditions, Environmental science & technology, 46, 5497–5503, doi:10.1021/es2040009, 2012.

Hensen, B., Lange, J., Jackisch, N., Zieger, F., Olsson, O., and Kümmerer, K.: Entry of biocides and their transformation products into groundwater via urban stormwater infiltration systems, Water research, 144, 413–423, doi:10.1016/j.watres.2018.07.046, 2018.

---

## Author Comment (AC3)

We would like to thank Adèle Bressy for the time she has taken to read our manuscript and her helpful comments to improve it. In the following section we are going to repeat the points brought up (in grey italic letters) and subsequently respond to them:

*Specific comments*
*Title: Perhaps specify in the title that it deals with stormwater* –
We will change the title to "Sources and pathways of biocides and their transformation products in urban stormwater infrastructures of a 2 ha urban district".

*Abstract: The last sentence of the abstract present obvious conclusions and not informative. It seems logical that by sampling in a targeted way, a better identification of the sources is obtained. Perhaps you can refine this conclusion and add some more concrete and precise results.*
Thank you for your suggestion. We will change the last sentence: The applied two-step approach determined sources and pathways of biocide and their TPs. This study contributes to expanding knowledge on their entry and distribution and thus eventually towards reducing emissions.

*Line 57: Please explain the choice of the TPs, why just these 3 compounds?*
These three compounds are commonly used as film protection products. They represent one herbicide, one algicide and one fungicide. Often, a combination of these and more compounds is used against algae and fungi growth (Sauer, 2017). All three compounds and the selected TPs have been part of previous studies on biocide runoff from facades, e.g. Burkhardt et al., 2011, Bollmann et al., 2016, Bollmann et al., 2017, Hensen et al., 2018, Paijens et al., 2021. For the quantification of TPs, an analytical standard needs to be available. Standards were available for the selected TPs. We will clarify our choice in an updated manuscript accordingly.

*Paragraph 2.1: the tow-step approach is well presented and convincing, but the long period between the first campaign (step 1 in 2015-2017) et the last one (step 2 in 2019-2020) raises the question of the comparability of the campaigns between them. Why did you not sample the swale system during the second campaign in 2019-2020 to verify the stability of the concentrations in the swale? Justify this point.*
Thanks for this comment. On a first glance, it really seemed obvious to continue swale sampling also in 2019-2020. However, as shown in Figure 4, biocide concentrations in the swale are highly variable and depended inter alia on event magnitude. Hence, we did not expect new findings from a renewed sampling campaign here. Instead, our objective was to concentrate on biocide sources and thereby limit the number of samples in an efficient campaign. We will stress this point in an updated manuscript.

*Line 117: you said that the last paint was in 2007 and after it is indicated that a façade was painted in 2018 (line 223). It is unclear.*
All buildings were painted last in 2007. However, there is one part of a façade that was re-painted due to restauration works in 2018. We will clarify this.

*Line 121: Is there always water in the swale or is it dry during dry weather?*
The swale is episodic, i.e. dry during dry weather. We will add this information.

*Line 153: how are sampled the roof, façade and pipe samples? Are they representative of the entire rain events? What about the first flush? You should add details about the sampling and its representativeness.*
All pipe samples (downpipes, street, drainage) were point samples during rain events and did not include first flush effects that might have shown higher concentrations. They are also not

representative of the entire rain event as no flow proportional samples were taken (see 3.4). We are aware of the concentration distribution of biocides during rain events and also of the first flush e.g. Bollmann et al., 2014. We will discuss this point in more detail in the manuscript.
Roof samples and facades samples were taken during artificial elution experiments as described in 2.3.2. We will rewrite this paragraph to describe the sampling in more detail.

*Line 153: how the water is sampled? You said during the sample, why not at the end of the sample?*
Water samples at rain downpipes were taken during the event. We chose to sample every event only once during the event because we had limited analytical capacity. See also comment above.

*Line 163-165: did you analyse the representativeness of the sampled events in relation to the classical pluviometry?*
Thank you for this idea. So far, we did not analyze the representativeness of the sampled events. In principle, sampling was only possible when there was water in the swale, which produced a bias towards large events. As shown in Fig. 4, we took samples during 3 of the 5 largest events in our measurement period. All sampled events were larger than 4mm/day. We will include a comparison to longer-term rainfall data and recurrence intervals in the updated manuscript. However, we will limit the validity of this analysis, since, as already described, the weather station is 5km away from the study area and there might be differences in local precipitation.

*Figure 4: I am not sure that this figure is really informative since we are not able to read the rainfall for each sampled event. Perhaps put it in supplementary materials*
This figure aims to show an overview of the chronology of the sampling. We will add the rainfall amount to the sampled events; see also comments to the referee #1 and #2.

*Line 176: why did you not test solar panel elution? Do you think that they could emit biocides?*
We are not aware of studies that found biocides used in solar panels, especially Diuron, Terbutryn and OIT measured in this study. We found very low Terbutryn concentrations and concentration of measured TPs in rain downpipes of houses with solar panels (Fig. 7). For this reason, we did not look for sources. We will clarify this point in the discussion.

*Line 177-178: n=1 seems insufficient to conclude.*
The limited significance of these samples will be discussed in the updated manuscript.

*Chemical analysis: I would recommend to present the analytical validations (as extraction recoveries) and the analytical uncertainties to validate the SPE extractions and the quantification.*
Thank you for your suggestion. We will add the extraction recoveries here. Recovery was determined by spiking water samples with 1 mg L$^{-1}$of analytical standard and was found to be 97.7 % (Diuron), 88.5 % (Terbutryn) and 93.5 % (OIT), 85.0 % (Diuron-desmethyl), 66.2 % (Terbumeton ), 50 % (Terbuthylazin-2-hydroxy) and 92 % (Terbutryn-desethyl) (Hensen et al., 2018).

*Paragraph 2.5: I am not really convinced by the methodology presented by this paragraph because the concentration used does not take into account the temporal evolution of emitted concentrations over time due to ageing or depletion of the stock in the material, or does not present an argument from the literature to overcome this. What verification have you implemented to justify the word "efficiently" in line 222. Justify the use of an average biocide concentrations to calculate BE. Moreover, why the used samples were not described in the 2.3 section? What is the number of the samples and the representativity? What is the sampling frequency? What is the variability of the measured concentrations? Does the concentration vary in time? Decrease? To prevent the reader's doubts, a part of the explanation from line 349 to 355 could be used in*

*the methodology presentation and the fact that the estimated BE will be compared to the literature.*

Thank you for this comment and your suggestions. We are aware that our approach is limited and only a rough estimate. We chose the word "efficiently" to stress that by only very few samples and little information on the building we obtained realistic estimations on biocide emissions over a two year time period. Sampling is described in 2.3.1 as part of sampling the rain downpipes. We describe concentrations in section 3.2.2. They vary for different events and rather decrease over time which compares to expectations. But the exact temporal evolution of concentrations cannot be determined based on only four and five point samples of events. We will modify this paragraph to clarify the limitation of our approach already in the method section.

*Paragraph 3.1: I am wondering if 4 sampled events are sufficient to assess the variability of the concentrations in the swale, especially since only one PNEC exceedance is observed to justify the continuation of the study. Why don't you continue to sample the swale in the second part of the study?*

See our answer to paragraph 2.1 above.

*Line 298: You said that you sampled an additional pipe (R4-2) because R4-1 exceeded R1 and R2 by an order of magnitude but you have no result for R1 and R2 before the first sampling of R4? I don't understand.*

Your observation is correct, thank you for finding this contradiction. We first sampled one pipe at one building (R4-1) and found concentrations that exceeded expected concentrations, because we did not expect roof areas as a biocide source. During the next event, we decided to sample another pipe at the same building (R4-2) to make sure there was no contamination in the first pipe. Additionally, we decided to sample one pipe at another house (R2). To confirm the low concentrations we sampled at a third building (R1). For comparison purposes we then sampled multiple events at all pipes. We will make this clear in an updated manuscript.

*Figure 8: Precise if it is mean or median values in the legend*

Shown are mean values. We will add this to the figure description.

*Line 320 and following: It is not clear if the difference of concentrations is due to the new paint or to the exposition.*

We will change this sentence to make it clear that the different concentrations are due to the new paint.

*Line 363/364: you explain that OIT was not detected due to its degradation in soil but for S9 (surface water pipe), water is not percolated through the soil? How do you explain to not found OIT in S9 samples?*

Concentrations of OIT at the facades were very low, i.e. 0.9-2.3ng/L. Hence, we did not expect to find OIT in the pipes. We will add this point to the manuscript.

*Line 413: you have to qualify this sentence because your method gave a rough and short term estimate (even if interesting). the comparison to the literature is only informative and does not bring proof of the accuracy of the evaluation, the initial stock of biocides not being necessarily the same.*

Thank you for pointing this out. We will clarify this statement. Samples at the rain down pipes just confirmed a continuous biocide leaching from the flat roof. Based on available measurements we did a rough estimation of the total long-term biocide leaching and a comparison of the obtained estimation with literature values to check, if we arrived at a realistic order of magnitude. We will clarify this both in the method and in the result section, see also our response to the comment on paragraph 2.5 above.

*Technical corrections*

*Figure 1: Step 2: perhaps precise "Elution and leaching test experiments"* - will be corrected.

*Line 97: two -s at "sselected"* - will be corrected.

*Line 114: The capital letter at "Area" is unnecessary* - will be corrected.

*Line 118: perhaps add a -s at "diverse use"?* - will be corrected.

*Line 134: I find that "surface water pipe" do not describe well the type of water sampled. It looks like surface water that has been sampled. Perhaps "surface runoff pipe" would be more meaningful*
Thank you for your suggestion, we will change the term accordingly.

*Figure 5: you could cut the ordinate-axis to better present the lowest concentrations.*
Thank you for your suggestion, we will change the figure accordingly.

*Line 246: space is missing between below and 4* - will be corrected.

*Line 245, 246 and 247: the sentence is not simple to understand*
We will change this sentence to make it clearer.

*Line 254: substanceS -*will be corrected.

*Line 296: perhaps add a coma after "in all rain downpipes" -*will be corrected.

*Figure 7 (d): perhaps precise "non sampled event". The use of one single scale is understandable but does not allow to read the concentrations for R1 and R2*
We will add non sampled events in the explanation of the figure.

*Line 321: due "to"? -*will be corrected.

*Line 579: "TEXTE"? -*will be changed.

References
Bollmann, U. E., Fernández-Calviño, D., Brandt, K. K., Storgaard, M. S., Sanderson, H., and Bester, K.: Biocide Runoff from Building Facades: Degradation Kinetics in Soil, Environmental science & technology, 51, 3694–3702, doi:10.1021/acs.est.6b05512, 2017.
Bollmann, U. E., Minelgaite, G., Schlüsener, M., Ternes, T., Vollertsen, J., and Bester, K.: Leaching of Terbutryn and Its Photodegradation Products from Artificial Walls under Natural Weather Conditions, Environmental science & technology, 50, 4289–4295, doi:10.1021/acs.est.5b05825, 2016.
Bollmann, U. E., Vollertsen, J., Carmeliet, J., and Bester, K.: Dynamics of biocide emissions from buildings in a suburban stormwater catchment - concentrations, mass loads and emission processes, Water research, 56, 66–76, doi:10.1016/j.watres.2014.02.033, 2014.
Burkhardt, M., Zuleeg, S., Vonbank, R., Schmid, P., Hean, S., Lamani, X., Bester, K., and Boller, M.: Leaching of additives from construction materials to urban storm water runoff, Water Science and Technology, 63, 1974–1982, doi:10.2166/wst.2011.128, 2011.

Hensen, B., Lange, J., Jackisch, N., Zieger, F., Olsson, O., and Kümmerer, K.: Entry of biocides and their transformation products into groundwater via urban stormwater infiltration systems, Water research, 144, 413–423, doi:10.1016/j.watres.2018.07.046, 2018.

Paijens, C., Bressy, A., Frère, B., Tedoldi, D., Mailler, R., Rocher, V., Neveu, P., and Moilleron, R.: Urban pathways of biocides towards surface waters during dry and wet weathers: Assessment at the Paris conurbation scale, Journal of hazardous materials, 402, 123765, doi:10.1016/j.jhazmat.2020.123765, 2021.

Sauer, F.: Microbicides in Coatings, 143 pp., 2017.

---

## Author Response (AR1)

**Please find the completed changes to the manuscript indented underneath each paragraph in the order of the referee comments. In the attached manuscript, changes are incorporated.**

**Answer to referee #1**

We would like to thank Ute Schoknecht for the time she has taken to read our manuscript and her helpful comments to improve it. In the following section, we are going to repeat the points brought up (in grey italic letters) and subsequently respond to them. We highlighted the changes in the updated manuscript in red:

*Specific comments*

*Line 48 Bollmann et al., 2016 and Bollmann et al., 2017b report on transformation of terbutryn and OIT on facades, please add these references here* – is added:

> Transformation of biocides can principally occur directly on treated objects (e.g. by photolysis on facades, Bollmann et al., 2016 and Bollmann et al., 2017b, Hensen et al., 2018) or along environmental pathways (e.g. in the soil, Bollmann et al., 2017a.).

*Line 50 231 days: this value was a result of Bollmann et al., 2017a-* is corrected:

> Degradation time of terbutryn in soil ranges between 10 days (Lechón et al., 1997) and 231 days (Bollmann et al., 2017a) depending on, among others, temperature, pH, organic and clay content.

*Line 57 There are numerous papers that report on transformation products of diuron, terbutryn and OIT. This statement could be related to façade coatings to avoid a long list of references. However, the papers of Bollmann et al., 2016and Bollmann et al., 2017b should be cited here at least. It may also be important that modern investigations on degradation products of diuron is usually limited to diuron-desmethyl although there are reports on a number of other transformation products (e.g. Jirkovský et al.: Photolysis of Diuron. Pesticide Science 50/1 (1997) 42-52 and other reports, see also Hensen et al. 2020). Possibly, other degradation products of diuron have been overlooked (not only in this study).*

We revised this paragraph so that its focus on selected TPs and on façade coatings becomes clearer. Regarding degradation products Diuron, we now include different relevant TPs. Thank you for your additional literature suggestions. We were not aware of the study by Jirkovský et al., 1997 and added this to the manuscript:

> Diuron, terbutryn and OIT used in façade coatings degrade to various transformation products (TPs, Hensen et al., 2020). Jirkovský et al., 1997 describe TPs of diuron formed by photolysis and Giacomazzi and Cochet, 2004 give an overview of all degradation pathways of Diuron. Bollmann et al., 2016 investigate photodegradation products formed at facades of Terbutryn and Bollmann et al., 2017b of OIT. Here, we focus on four commonly investigated TPs of diuron and terbutryn originating at facades (diuron-desmethyl, terbuthylazin-2-hydroxy, terbutryn-desethyl and terbumeton). Terbuthylazin-2-hydroxy and terbutryn-desethyl are formed by photolysis or biodegradation (Burkhardt et al., 2012; Bollmann et al., 2016; Bollmann et al., 2017a; Hensen et al., 2018). In a leaching study under natural weather conditions, Bollmann et al. (2016) found terbuthylazin-2-hydroxy, terbutryn-desethyl and terbumeton in render and in leachate. Terbumeton is a photo degradation product that tends to remain on facades (Bollmann et al., 2017a). Terbutryn-desethyl, terbuthylazin-2-hydroxy and terbumeton were classified as probably toxic (Hensen et al., 2020). Diuron-desmethyl was identified as a photo TP (Burkhardt et al., 2012; Hensen et al., 2018) and is possibly also formed by microorganisms in soil (Hensen et al., 2018). Diuron-desmethyl was detected in urban runoff by various studies (Wittmer et al., 2010; Reemtsma et al., 2013; Hensen et al., 2018). In a field experiment only 0.4 % of the diuron losses were made of diuron-desmethyl (Burkhardt et al., 2012). Moschet et al. (2014)

confirmed diuron-desmethyl in rivers in Switzerland at concentrations ranging from 10 to 22 ng L$^{-1}$. Diuron-desmethyl was classified as most probably toxic or probably toxic (Hensen et al., 2020).

*Line 69 Biocides and their TPs can enter the environment only in case of driving rain to the surface (not generally during rain events).* - We clarified our statement:

Biocides and their TPs are washed off from facades and enter the environment due to wind driven rain to the facade.

*Line 70 Please explain how the elution experiments were performed on roof materials from house 4 – especially in case of horizontal orientation.*

This comment refers to line 170. We included more details on the elution experiments as follows:

Most roof materials (i.e. roofing foils, roof access, roof cladding, elevator shaft foil and grass foils) were tested where their orientation was vertical, for example, around vertical orientated pipes or shafts. Railings were accessible from all sides allowing us to conduct elution experiments by setting a container underneath them. At the railings there were certain limitations regarding area poured with water so this might not be comparable to experiments at the facades. We dismantled parts of the wooden terraces to access the substructure and set a container underneath the wooden bars. This way leaching experiments on a horizontal surface the same way as on the facades were performed.

*Line 196 Please add information on the recovery of the SPE procedure for the analytes.*

Recovery was determined by spiking water samples with 1 mg L$^{-1}$of analytical standard and was found to be 97.7 % (Diuron), 88.5 % (Terbutryn) and 93.5 % (OIT), 85.0 % (Diuron-desmethyl), 66.2 % (Terbumeton ), 50 % (Terbuthylazin-2-hydroxy) and 92 % (Terbutryn-desethyl) (Hensen et al., 2018).

*Line 230 Estimation of net BE: The estimation of net biocide emissions according the given formula cannot be correct. Several reasons why this is incorrect are discussed later in the text (Line 349). Please change wording under 2.5 to clarify that this calculation is a rough estimate with certain reservations.*

We are aware that our estimation has many limitations and only gives a very rough estimate. We clarified this as follows:

Sentences added to 2.5:

Note that BE is only a rough estimation with various limitations discussed in 3.2.3. These include applied literature values for initial amount of biocides and paints, no consideration of dry and wet periods or wind driven rain, material aging and limited sampling. We compared the estimated BE with literature values to determine whether estimations are feasible.

Sentence changed in 3.2.3

Regardless of these uncertainties, we arrived at a realistic order of magnitude why we consider our approach promising for an initial estimation of relevant biocide sources by a limited number of samples.

*Line 255 Differences in substance patterns are probably also caused by different intensity of UV radiation.*

We added this as a possible explanation also here. This nicely fits into the paper, since we have also mentioned the impact of UV radiation in the introduction and in section 3.2.1:

Differences in types and concentrations of detected substances between locations and events might be due to different sources (e.g. newly painted facades), different intensities of UV radiation and different precipitation amounts and intensities that affect biocide emissions (Paijens et al., 2019)

*Line 277 Different patterns of transformation products depending on different pigments were observed by Urbanczyk et al. 2019 (Influence of pigments on phototransformation of biocides in paints. Journal of Hazardous Materials 364 (2019) 125-133).*

Thank you for your remark and reference. We added this fact as an additional explanation:

Urbanczyk et al., 2019 found differences in pigments contained in paints and renders influencing formation of TPs.

*Line 435: Missing biocides in the samples is not necessarily explained by former wash-off. Water solubility of most transformation products is probably higher than water solubility of the biocides. Therefore, the TPs should be washed off easier than the biocides. Probably, biocides that were available on the surface were almost completely transformed. It cannot be excluded, that biocides are still present in deeper layers of the materials that were not reached during the very short elution experiment.*

Thank you for clarifying. We added this explanation to the discussion on façade (Section 3.2.1). We also shortened the original sentence:

Original sentence shortened, line 435:
Some facades showed only TPs but no biocides.
Added to 3.2.1:
Higher water solubility of most TPs compared to biocides might lead to more wash-off of TPs and thus easier detection of TPs. Additionally, biocides might still be present in deeper layers of the facade while degraded on the surface (Uhlig et al., 2019).

*Line 444: For environmental risk assessments it is urgently required whether PNEC values are occasionally or permanently exceeded. The data for the swale indicate that the PNEC values for diuron and terbutryn were exceeded in one out of four samples from the swale. Please clarify this statement.*

Thank you for pointing this out. We did not aim for a complete environmental risk assessment and clarified our statement by inserting the word "potential". We are aware that a limited number of measurements cannot give information about long-term environmental risk. However, we wanted to stress the fact that biocide pollution remains an issue after more than a decade after construction has ended:

For surface water, our study confirmed the potential environmental risk of biocide use, since concentrations at the outlet of our urban catchment exceeded PNEC values at one event.

*Technical Corrections*

*Title : please add a blank between '2' and 'ha'* – corrected:
Sources and pathways of biocides and their transformation products in urban stormwater infrastructure of a 2 ha urban district

*Line 14 use capital letters for Central and Northern Europe (also in the following text)* – corrected:
Line 14:
Sampling utilizes existing urban water infrastructure representative for decentralized storm water management in Central and Northern Europe and and applies a two-step approach to (a) determine the occurrence of biocides above water quality limits (i.e. predicted no effect concentration, PNEC) and (b) identify source areas and characterize entry pathways into surface- and groundwater.
Line 122:
Similar development areas exist in other parts of the city and are typical for modern architecture in Central and Northern Europe.

*Line 64 :please delete either 'and' or 'but'* – *corrected:*
Diuron-desmethyl was identified as a photo TP (Burkhardt et al., 2012; Hensen et al., 2018) and is possibly also formed by microorganisms in soil (Hensen et al., 2018).

*Line 95 :Please check the number of samples (52). The number of samples described in Table 1 amounts to 49. 3 samples from artificial experiments on facades and 20 samples from artificial*

*experiments on roof materials from house 4 and x samples from a leaching test on the wooden terrace are mentioned in the text. -*
*Possibly, the origin of the samples can be mentioned here (collected in the swale, rain downpipes and drainage pipe; from elution experiments on facades and roof materials from house 4 and a leaching test on the wooden terrace).*
We updated the table and made it clear.

Using a stepwise approach and making use of existing urban water infrastructure, this study characterizes the environmental hazard of urban biocide use with only a small number of samples (n = 60), thus limiting laboratory expenses.

Table 1: Overview of samples taken.

| STEP | Sample type | Sample location | Name | Events sampled | Number of samples | Additional duplicates |
|---|---|---|---|---|---|---|
| 1 | Event samples | Swale | swale | 4 | 4 | 1 |
| 2 Part 1 | Event samples | Rain downpipes | R1 | 3 | 3 | 2 |
| | | | R2 | 4 | 4 | 1 |
| | | | R4-P1 | 5 | 5 | 4 |
| | | | R4-P2 | 4 | 4 | 3 |
| | Elution experiments | Facades | F2 | | 2 | |
| | | | F4 | | 2 | |
| | | | F5 | | 2 | |
| | | Roof materials | | | 14 | |
| | | Newly painted roof facade | | | 4 | |
| | | Old roof facade | | | 2 | |
| | Leaching test | Wooden terrace | | | 2 | |
| 2 Part 2 | Event samples | Drainage pipes | S9 | 4 | 4 | 2 |
| | | | P10 | 4 | 4 | 2 |
| | | | P11 | 4 | 4 | 2 |
| | TOTAL | | | | 60 | 17 |

*Line 97 : please correct: 'selected' –* corrected:

Thereby, potential sources of biocides and TPs are identified by sampling at selected points of urban infrastructure, e.g. rain downpipes of flat roofs and by artificial experiments on facades and roof materials.

*Figure 1 please correct: Step 2 – Part 1 Identify source areas (instead of 'sources')* –corrected
*Method: the phrase 'elution experiments at selected infrastructures' would facilitate understanding the different methods mentioned here* –corrected:

Figure 1 changed accordingly:

| **Step 1: Determine biocides and TPs in aquatic systems in the study area** | |
|---|---|
| Urban water infrastructure: | *Method:* |
|    Swale or swale-trench system
   Retention ponds | *Sampling at the catchment outlet* |
| Test criterion: measured biocide concentrations higher than available guidance values (e.g. PNEC) | |

*If measured concentrations higher than defined test criterion*

| **Step 2: Identify sources and characterize pathways to possibly reduce biocide entry** | |
|---|---|
| **Step 2 – Part 1 Identify source areas** | **Step 2 – Part 2 Characterize pathways** |
| Urban water infrastructure:
   Rain downpipes
   Flat roofs
   …
Method:
*Elution and leaching experiments at selected infrastructures*
*Sampling at selected infrastructures*
*Estimation of biocide emissions* | Urban water infrastructure:
   Drainage pipes
   Runoff from paved or unsealed surfaces
   …
Method:
*Sampling at selected infrastructures* |

*Table 1 The information '(Duplicates >1)' seems to be unnecessary and rather confusing.*
Thank you for your suggestion. Indeed, this might be a bit confusing. We changed the table 1, see above changes in table 1 at comment line 95.

*Line 246: please add a blank between 'below' and '4'* - corrected
Sentence was changed according to referee #3.
In the same study, diuron showed maximum concentrations of up to 0.6 µg L$^{-1}$ and OIT up to 60 ng L$^{-1}$. Gasperi et al., 2013 analyzed storm water at the outlet of three catchments in Paris, Nantes and near Lyon. Mean concentrations were 2 µg L$^{-1}$ for diuron and OIT concentrations remained below 4 ng L$^{-1}$.

*Line 250 : please add 'in samples' after 'desmethyl'* – corrected:
The TPs terbuthylazin-2-hydroxy and terbutryn-desethyl were detected in samples of the first two and the fourth event, diuron-desmethyl in samples of the second and third event.

*Line 253: please delete 'detect'* – deleted:
Hensen et al. (2018) found concentrations of up to 23 ng L$^{-1}$ for terbuthylazin-2-hydroxy, 73 ng L$^{-1}$ for terbutryn-desethyl and 2 ng L$^{-1}$ for diuron-desmethyl in a swale-trench system located 2.8 km southwest.

*Line 432: please correct: 'systems'* – corrected:
Receiving urban infiltration systems (e.g. swales, swale-trench systems, retention ponds) at the outlet of a catchment generally provide an integrated signal of the aquatic system of a larger area.

*Line 500: please add a link*
This refers to line 501 where a link is missing. We added the link accordingly (https://echa.europa.eu/de/information-on-chemicals/biocidal-active-substances).
ECHA: List of biocidal active substances, https://echa.europa.eu/de/information-on-chemicals/biocidal-active-substances, 2007-2020.

Line 510 please add a link –added: (https://www.thesourcemagazine.org/urban-groundwater-
mobilising-stakeholders-to-improve-monitoring/ )

Foster, S. and Cogu, R. C.: Urban Groundwater -- mobilising stakeholders to improve monitoring, The Source, https://www.thesourcemagazine.org/urban-groundwater-mobilising-stakeholders-to-improve-monitoring/, 2019.

In References, 3 references added as suggested from referee #1, former line 57; referee #1, former line 277 and referee #1, former line 435:

Jirkovský, J., Faure, V., and Boule, P.: Photolysis of Diuron, Pestic. Sci., 50, 42–52, doi:10.1002/(SICI)1096-9063(199705)50:1<42:AID-PS557>3.0.CO;2-W, 1997.

Uhlig, S., Colson, B., and Schoknecht, U.: A mathematical approach for the analysis of data obtained from the monitoring of biocides leached from treated materials exposed to outdoor conditions, Chemosphere, 228, 271–277, doi:10.1016/j.chemosphere.2019.04.102, 2019.

Urbanczyk, M. M., Bester, K., Borho, N., Schoknecht, U., and Bollmann, U. E.: Influence of pigments on phototransformation of biocides in paints, Journal of hazardous materials, 364, 125–133, doi:10.1016/j.jhazmat.2018.10.018, 2019.

**References**

Bollmann, U. E., Fernández-Calviño, D., Brandt, K. K., Storgaard, M. S., Sanderson, H., and Bester, K.: Biocide Runoff from Building Facades: Degradation Kinetics in Soil, Environmental science & technology, 51, 3694–3702, doi:10.1021/acs.est.6b05512, 2017a.

Bollmann, U. E., Minelgaite, G., Schlüsener, M., Ternes, T., Vollertsen, J., and Bester, K.: Leaching of Terbutryn and Its Photodegradation Products from Artificial Walls under Natural Weather Conditions, Environmental science & technology, 50, 4289–4295, doi:10.1021/acs.est.5b05825, 2016.

Bollmann, U. E., Minelgaite, G., Schlüsener, M., Ternes, T. A., Vollertsen, J., and Bester, K.: Photodegradation of octylisothiazolinone and semi-field emissions from facade coatings, Scientific reports, 7, 41501, doi:10.1038/srep41501, 2017b.

Giacomazzi, S. and Cochet, N.: Environmental impact of diuron transformation: a review, Chemosphere, 56, 1021–1032, doi:10.1016/j.chemosphere.2004.04.061, 2004.

Hensen, B., Lange, J., Jackisch, N., Zieger, F., Olsson, O., and Kümmerer, K.: Entry of biocides and their transformation products into groundwater via urban stormwater infiltration systems, Water research, 144, 413–423, doi:10.1016/j.watres.2018.07.046, 2018.

Jirkovský, J., Faure, V., and Boule, P.: Photolysis of Diuron, Pestic. Sci., 50, 42–52, doi:10.1002/(SICI)1096-9063(199705)50:1<42:AID-PS557>3.0.CO;2-W, 1997.

**Answer to referee #2**

We would like to thank the anonymous referee for the time he/she has taken to read our manuscript and his/her helpful comments to improve it and gain clarity. In the following section, we are going to repeat the points brought up (in grey italic letters) and subsequently respond to them. We highlighted the changes in the updated manuscript in red:

*Abstract*
*Line 18-20: This sentence is very difficult to read please divide it into two sentences.* – We changed this sentence accordingly.

> We confirmed expected sources, i.e. facades. Sampling of rain downpipes from flat roofs identified additional sources of all biocides and two TPs of terbutryn and one TP of diuron.

*Line 23-24: Revise the sentence sintaxis, very difficult to read-*– We changed this sentence:

> One of the pipes collecting infiltrated water through soil concentration showed highest concentrations of terbutryn and two of its TPs (terbutryn-desethyl and terbuthylazin-2-hydroxy). This suggests a high leaching potential of terbutryn.

*Line 25: Delete "for" after "allows".*– corrected:

> Sentence changed according to referee #3:
> The applied two-step approach determined sources and pathways of biocide and their TPs. This study contributes to expanding knowledge on their entry and distribution and thus eventually towards reducing emissions.

*Introduction*

*Line 40-44: Please divide this sentences into two or serveral sentences, is to difficult to read.*
We revised these sentences to make them clearer:

> Use of terbutryn and OIT is additionally legal for fibre, leather, rubber and polymerized materials preservatives (Product-Type 09). In addition, OIT is open for use in preservatives for products during storage (Product-Type 06), wood preservatives (Product-Type 08), preservatives for liquid-cooling and processing systems (Product-Type 11) and working or cutting fluid preservatives (Product-Type 13) (ECHA, 2007-2020).

*Line 46-47: This information is already included in Table 3, please delete.* –corrected:

> Sentence deleted.

*Line 50: Delete "for example"* - We changed this to "among others" as we mention the most common influences.

> Degradation time of terbutryn in soil ranges between 10 days (Lechón et al., 1997) and 231 days (Bollmann et al., 2017a) depending on, among others, temperature, pH, organic and clay content.

*Line 57: Delete the comma, instead place TPs in parenthesis.* –corrected

> Sentence changed:
> Diuron, terbutryn and OIT used in façade coatings degrade to various transformation products (TPs, Hensen et al., 2020).

*Line 57-58: This sentence should be at the end of the paragraph.* – We revised this paragraph:

> Paragraph changed, according to referee #1:
> Diuron, terbutryn and OIT used in façade coatings degrade to various transformation products (TPs, Hensen et al., 2020). Jirkovský et al., 1997 describe TPs of diuron formed by photolysis and Giacomazzi and Cochet, 2004 give an overview of all degradation pathways of Diuron. Bollmann et al., 2016 investigate photodegradation products formed at facades of Terbutryn and Bollmann et al., 2017b of OIT. Here, we focus on four commonly investigated TPs of diuron and terbutryn originating at facades (diuron-desmethyl, terbuthylazin-2-hydroxy, terbutryn-desethyl and terbumeton).

Terbuthylazin-2-hydroxy and terbutryn-desethyl are formed by photolysis or biodegradation (Burkhardt et al., 2012; Bollmann et al., 2016; Bollmann et al., 2017a; Hensen et al., 2018).

*Line 77: Geometry?* – *We added this influence and referred to Burkhardt et al., 2012:*

Release of biocides from facades is controlled by temperature, time between rain events, their extent, wind, UV exposure, biocide characteristics and properties of paint and renders used (Paijens et al., 2019), as well as architectural design and geometry (Burkhardt et al., 2012).

*Line 81: "Studies have confirmed…"* – corrected:

Studies have confirmed general biocide emissions from larger heterogeneous residential areas in storm water channels of separated sewer systems (Bollmann et al., 2014b; Wicke et al., 2015).

*Line 86: Please delete the comma and better add swale-trench system in parenthesis.*

We corrected the comma. We are aware that various terms exist for urban storm water infiltration systems. We defined the term swale-trench system more clearly by referring to the study of Hensen et al., 2018:

In another study, Hensen et al. (2018) investigated biocide emission from two small urban catchments with sizes of 2.95 ha and 8,047 m², but focused on the receiving parts of the water infrastructure (swale-trench system).

*Line 88-89: This sentence is very difficult to read.* – We improved this sentence:

Existing studies often do not systematically follow the fate of biocides including TPs from source to sink. This is especially the case for urban districts with buildings' ages of 10 years or more.

*Methods*
*Line 109-110:    MEC/PNEC where chosen for what? Which criterion? Relevance threshold?*

In a first step, the objective was to determine the relevance of biocides in our study area. We chose MEC/PNEC >1 as a common threshold for environmental risk assessment. We are aware that our study is not a complete environmental risk assessment, rather a "potential" risk assessment, hence we added "potential" in the updated manuscript. Our objective rather was to have a defined starting point for further investigations. See also our answers to referee #1 Ute Schoknecht.

For surface water, our study confirmed the potential environmental risk of biocide use, since concentrations at the outlet of our urban catchment exceeded PNEC values at one event.

*Line 115: Please add coordinates* - We added coordinates: 47° 59N  7° 51E.

The study area is located in the city of Freiburg in south-western Germany (47° 59N, 7° 51E).

*Line 126-128:    Please add here the total facade area if possible, or size of the buildings and roof top total area approx. Its important to have an idea of the biocide loads from each of the buildings or from the total building complex.*

Thank you for your suggestion. We added information on the geometry of the buildings:

Footprints of buildings vary from 624 to 214m² (houses 1 to 8) with an approximate height of 13 m. Estimated facade areas covered by paints and renders are 634m² to 296m² for houses 1 to 8. All houses were constructed at the same time and thus exposed to identical weather conditions over the years. At two houses (3, 4) used render contains diuron and OIT according to inhabitants and invoices of construction work. For the other houses, used paint or renders could not be identified.
Houses 1, 2 and 4 all have flat roofs that are mostly covered by extensive greening. House 1 additionally has small roof top terraces with an area of 44 m². House 2 has an extensive green area with solar panels but no roof top terrace. House 4 contains two larger roof top terraces with approximate areas of 63 m² and 96 m².

*Line 131-134: Please add the pipeline/drainage material-*

Information was added to line 143:

The material of these pipes is polyvinyl chloride.

*Line 152-155: Please add the total amount of samples within the test period*

We changed Table 1 so that the number of samples becomes clearer. See also our answers to referee #1 Ute Schoknecht.

See changes to Table 1 above, answer to referee #1.

*Line 180: Is this leaching test out of norm/standard (i.e DSLT) or it is a self fabricated test? If it is, please argue why you do the leaching test that way.*

The aim here was to account if any leaching takes place at all. That is why we used a self-fabricated test. The first leaching test did not show any biocide concentrations. Further elution experiments at other parts of the wooden terrace confirmed that this was not a biocide source. We discussed this in the updated manuscript as follows:

An additional self-fabricated leaching test was performed on the wooden terrace. The aim of this test was to determine if any leaching takes place.

*Line 195: Instead of "measurement" use "analysis".–*corrected:

Analysis of environmental samples were performed with a Triple Quadrupole (Agilent Technologies, 1200 Infinity LC-System and 6430 Triple Quad, Waldbronn, Germany) with ESI in positive mode.

*Table 3: Please add water solubility, half-life time, molecular mass and lethal dose. –* added as follows:

**Table 3: Overview of analyzed substance. According to (a) Bollmann et al., 2016, (b) Bollmann et al. 2017, (c) Hensen et al. (2018) and (d) Paijens et al. (2019).**

| Substance | Molecular Formula | Chemical Structure | Log $K_{OW}$ at pH 7 | CAS-No. | PNEC [µg $L^{-1}$] | Water solubility [mg $L^{-1}$] | Half-life in soil [d] | Molar mass [g $mol^{-1}$] | EC 50 daphnia magna 48h [mg $L^{-1}$] |
|---|---|---|---|---|---|---|---|---|---|
| Diuron | $C_9H_{10}Cl_2N_2O$ | | 2.71 - 2.85 [d] | 330-54-1 | 0.02 [d] | 102 [d] | >2500 [b] | 233.1 [d] | 5.7 [d] |
| Terbutryn | $C_{10}H_{19}N_5S$ | | 3.65 [d] | 886-50-0 | 0.034 [d] | 42 [d] | 231 [b] | 241.4 [d] | 2.6 [d] |
| Octylisothiazolinone (OIT) | $C_{11}H_{19}NOS$ | | 2.45 - 2.61 [d] | 26530-20-1 | 0.013 [d] | 309 [d] | 9.3 [b] | 213.3 [d] | 0.32 [d] |
| Diuron-desmethyl (diuron TP-219) | $C_8H_8Cl_2N_2O$ | | | 3567-62-2 | | | | 219 [c] | |

| Compound | Formula | | | | | |
|---|---|---|---|---|---|---|
| Terbuthylazin-2-hydroxy (terbutryn TP-212) | $C_9H_{17}N_5O$ | | 1.5[a] | 66753-07-9 | 906[a] | 212.2[a] |

| Compound | Formula | | | | | |
|---|---|---|---|---|---|---|
| Terbutryn-desethyl(terbutryn TP-214) | $C_8H_{15}N_5S$ | | 2.7[a] | 30125-65-6 | 174[a] | 214.1[c] |

| Compound | Formula | | | | | |
|---|---|---|---|---|---|---|
| Terbumeton (terbutryn TP-226) | $C_9H_{19}N_5O$ | | 3.6[a] | 33693-04-8 | 73[a] | 226.2[a] |

Results

*Line 245: "There, diuron showed maximum concentrations of…"*– corrected:

Sentence changed according to referee #3.:

In the same study, diuron showed maximum concentrations of up to 0.6 µg L$^{-1}$ and OIT up to 60 ng L$^{-1}$.

*Line 255: Please add weather data elsewhere in studied area/sampling site (methods section). Here you argue about weather conditons in the area but there is no information of it prior this argumentation.*

We did not have a weather station in the immediate district, but relied on a weather station about 5km away from the study area, see 2.3.1. In Fig. 4 we used rainfall data to illustrate rainfall magnitudes during the sampled events. We added the amount of precipitation in an updated figure:

Updated Fig. 4:

[Figure]

**Figure 4: Daily precipitation about 5 km away from study site between 2015 and 2020. Sampled events are marked according to the two-step approach. Colors correspond to steps. Precipitation data taken from Deutscher Wetterdienst, Station Freiburg.**

*Section 3.2.1:    Does the impinged water volumes have an influence in the leaching concentrations? All the facades received the same amount of water? Are collected runoffs in the same order? It is important to mention this since the leaching amount of substances is also dependant on the contact water volume. Higher the runoff volume, higher the substance load.*
*Please mention in this section something about the contact water volume, it is an important parameter into consideration when talking about substance leaching of facades. Consider biocide loads (mg/m² or µg/m²) in this section, since this measurement is important for environmental evaluation properties of any construction site.*

Thank you for your comment. We are aware that the impinged water volume has an influence on the leaching concentrations. We conducted the elution experiments as similar as possible to reduce such influences. Collected runoff volumes were in the same order of magnitude, about 1L.
We sprinkled about 1L across 0.25m² and collected the entire runoff (see 2.3.2). We repeated these experiments twice at each investigated façade and found similar concentrations in the obtained duplicates. We clarified this point in the discussion of the updated manuscript and stressed that the results should not be evaluated quantitatively but rather qualitatively in a sense that a specific

biocide was detected or not. This also due to the fact that information on initial biocide loads could not be determined for all buildings (see 2.2).

We added the following sentences to 3.2.1:

We conducted the elution experiment twice at different parts of the facades and found similar concentrations in the obtained duplicates. Due to our simple experimental approach and missing information about initial biocide loads, we will focus on a qualitative evaluation of the results.

*Line 421: Please delete "Again"`*–corrected:

We used existing urban infrastructure, in this case the collection of areal infiltration by a drainage system on top of an underground parking garage.

**References**

Burkhardt, M., Zuleeg, S., Vonbank, R., Bester, K., Carmeliet, J., Boller, M., and Wangler, T.: Leaching of biocides from façades under natural weather conditions, Environmental science & technology, 46, 5497–5503, doi:10.1021/es2040009, 2012.

Hensen, B., Lange, J., Jackisch, N., Zieger, F., Olsson, O., and Kümmerer, K.: Entry of biocides and their transformation products into groundwater via urban stormwater infiltration systems, Water research, 144, 413–423, doi:10.1016/j.watres.2018.07.046, 2018.

**Answer to referee #3**

We would like to thank Adèle Bressy for the time she has taken to read our manuscript and her helpful comments to improve it. In the following section, we are going to repeat the points brought up (in grey italic letters) and subsequently respond to them. We highlighted the changes in the updated manuscript in red:

*Specific comments*
*Title: Perhaps specify in the title that it deals with stormwater* –
We changed the title to:

> Sources and pathways of biocides and their transformation products in urban stormwater infrastructure of a 2 ha urban district

*Abstract: The last sentence of the abstract present obvious conclusions and not informative. It seems logical that by sampling in a targeted way, a better identification of the sources is obtained. Perhaps you can refine this conclusion and add some more concrete and precise results.*
Thank you for your suggestion. We changed the last sentence:

> The applied two-step approach determined sources and pathways of biocide and their TPs. This study contributes to expanding knowledge on their entry and distribution and thus eventually towards reducing emissions.

*Line 57: Please explain the choice of the TPs, why just these 3 compounds?*
These three compounds are commonly used as film protection products. They represent one herbicide, one algicide and one fungicide. Often, a combination of these and more compounds is used against algae and fungi growth (Sauer, 2017). All three compounds and the selected TPs have been part of previous studies on biocide runoff from facades, e.g. Burkhardt et al., 2011, Bollmann et al., 2016, Bollmann et al., 2017, Hensen et al., 2018, Paijens et al., 2021. For the quantification of TPs, an analytical standard needs to be available. Standards were available for the selected TPs.

> We added a sentence to clarify substance choice after introducing the 3 selected biocides:
> Often, a combination of these three and more compounds is used against algae and fungi growth (Sauer, 2017).
> We also revised the paragraph of line 57, see changes to referee #1:
> Diuron, terbutryn and OIT used in façade coatings degrade to various transformation products (TPs, Hensen et al., 2020). Jirkovský et al., 1997 describe TPs of diuron formed by photolysis and Giacomazzi and Cochet, 2004 give an overview of all degradation pathways of Diuron. Bollmann et al., 2016 investigate photodegradation products formed at facades of Terbutryn and Bollmann et al., 2017b of OIT. Here, we focus on four commonly investigated TPs of diuron and terbutryn originating at facades (diuron-desmethyl, terbuthylazin-2-hydroxy, terbutryn-desethyl and terbumeton). Terbuthylazin-2-hydroxy and terbutryn-desethyl are formed by photolysis or biodegradation (Burkhardt et al., 2012; Bollmann et al., 2016; Bollmann et al., 2017a; Hensen et al., 2018).

*Paragraph 2.1: the tow-step approach is well presented and convincing, but the long period between the first campaign (step 1 in 2015-2017) et the last one (step 2 in 2019-2020) raises the question of the comparability of the campaigns between them. Why did you not sample the swale system during the second campaign in 2019-2020 to verify the stability of the concentrations in the swale? Justify this point.*
Thanks for this comment. On a first glance, it really seemed obvious to continue swale sampling also in 2019-2020. However, as shown in Figure 4, biocide concentrations in the swale are highly variable and depended inter alia on event magnitude. Hence, we did not expect new findings from a renewed sampling campaign here. Instead, our objective was to concentrate on biocide sources and thereby limit the number of samples in an efficient campaign. We stressed this point in the updated manuscript by adding a sentence to section 3.1:

Due to dependencies on rain event magnitudes and thus high variability of detected concentrations in the swale, we did not expect new findings from a renewed sampling campaign in 2019-2020 and thus focused on sources of biocides in the next step.

*Line 117: you said that the last paint was in 2007 and after it is indicated that a façade was painted in 2018 (line 223). It is unclear.*

All buildings were painted last in 2007. However, there is one part of a façade that was re-painted due to restauration works in 2018. We clarified this and changed the sentence:

The modern four-story houses with thermal insulation composite systems were lastly painted in 2007 except for one small part of a roof that was re-painted due to restauration works in 2018. We obtained this information from a survey among residents and architects.

*Line 121: Is there always water in the swale or is it dry during dry weather?*

The swale is episodic, i.e. dry during dry weather. We added this information:

The study area consists of eight houses connected to a separated sewer system that ends up in a swale that is dry during dry weather (Fig. 2).

*Line 153: how are sampled the roof, façade and pipe samples? Are they representative of the entire rain events? What about the first flush? You should add details about the sampling and its representativeness.*

All pipe samples (downpipes, street, drainage) were point samples during rain events and did not include first flush effects that might have shown higher concentrations. They are also not representative of the entire rain event as no flow proportional samples were taken (see 3.4). We are aware of the concentration distribution of biocides during rain events and also of the first flush e.g. Bollmann et al., 2014. Roof samples and facades samples were taken during artificial elution experiments as described in 2.3.2.

In Section 2.3.1, sentence changed to:
Samples were taken as grab samples during the events and are not representative of the entire rain event.
In Section 3.4 we added the following sentence:
Concentrations of biocides can vary within an event including first flush dynamics (Bollmann et al., 2014b).

*Line 153: how the water is sampled? You said during the sample, why not at the end of the sample?*

Water samples at rain downpipes were taken during the event. We chose to sample every event only once with point sampling during the event to reduce analytical efforts.

See 3.4.:
With a limited number of samples and analyzed substances especially small-scale districts can be characterized regarding their potential risk of biocide emissions.

*Line 163-165: did you analyse the representativeness of the sampled events in relation to the classical pluviometry?*

Thank you for this idea. So far, we did not analyze the representativeness of the sampled events. In principle, sampling was only possible when there was water in the swale, which produced a bias towards large events. As shown in Fig. 4, we took samples during 3 of the 5 largest events in our measurement period. All sampled events were larger than 4mm/day. We included a comparison to longer-term rainfall data and recurrence intervals in the updated manuscript. However, we also stressed limited validity of this analysis, since our weather station is 5km away from the study area and there might be differences in local precipitation.

In Section 2.3.1 we added the following sentence:
We determined the representativeness of the sampled rain events by comparisons to a 30-year period of rainfall data.
In section 3.1 we added the following sentences:
It corresponded to the 5th largest daily rainfall in 1990-2020 period. All our sampled events were larger than 4mm per day although this only applied to 39% of the events in the 1990-2020 period. Two of the four sampled events in the swale exceeded 30 mm per day which was the case for only 1% of the 1990-

2020 events. These findings suggest a bias towards larger events in our sampling. However, this analysis is limited since the weather station providing long term rainfall data is located 5 km away from the investigated site and there might be differences in local precipitation.

*Figure 4: I am not sure that this figure is really informative since we are not able to read the rainfall for each sampled event. Perhaps put it in supplementary materials*
This figure aims to show an overview of the chronology of the sampling. We added the rainfall amounts to the sampled events; see also comments to the referee #1 and #2.

*Line 176: why did you not test solar panel elution? Do you think that they could emit biocides?*
We are not aware of studies that found biocides used in solar panels, especially Diuron, Terbutryn and OIT measured in this study. We found very low Terbutryn concentrations and concentration of measured TPs in rain downpipes of houses with solar panels (Fig. 7). For this reason, we did not look for sources. We clarified this point in the discussion and added the following sentence:
To the best of our knowledge, we are not aware of studies that found biocides emitted from solar panels that are installed on the roof of house 2 where we detected low Terbutryn concentrations.

*Line 177-178: n=1 seems insufficient to conclude.*
The limited significance of these samples is now discussed in the updated manuscript.
In Section 3.2.2 we revised the corresponding paragraph:
Leaching tests of the wooden roof terrace taken from house 4 showed no biocides or TPs present in the extraction solution. Elution experiments of the wooden roof terrace showed no concentrations and thus confirmed findings of the self-fabricated leaching test. Elution experiments of various roof materials showed very low concentrations of terbutryn ($<1$ ng L$^{-1}$) (Fig. 8a), while OIT was found in the railing, in the roof foil and in the roof access (max. 12 ng L$^{-1}$). The significance of the results of the elution experiments of roof materials is limited due to the fact that three materials were only sampled and measured once (roof access, roof cladding, elevator shaft foil and grass foil). Still, these low concentrations did not suggest a primary source as it was indicated by the findings in the rain downpipes. However, elution experiments at parts of the inner roof facade yielded very high concentrations (2.7 µg L$^{-1}$ diuron, 2.6 µg L$^{-1}$ diuron-desmethyl and 1.9 µg L$^{-1}$ OIT, Fig. 8b).

*Chemical analysis: I would recommend to present the analytical validations (as extraction recoveries) and the analytical uncertainties to validate the SPE extractions and the quantification.*
Thank you for your suggestion. We now added the extraction recoveries here.
Changed according to referee #1:
Recovery was determined by spiking water samples with 1 mg L$^{-1}$ of analytical standard and was found to be 97.7 % (Diuron), 88.5 % (Terbutryn) and 93.5 % (OIT), 85.0 % (Diuron-desmethyl), 66.2 % (Terbumeton ), 50 % (Terbuthylazin-2-hydroxy) and 92 % (Terbutryn-desethyl) (Hensen et al., 2018).

*Paragraph 2.5: I am not really convinced by the methodology presented by this paragraph because the concentration used does not take into account the temporal evolution of emitted concentrations over time due to ageing or depletion of the stock in the material, or does not present an argument from the literature to overcome this. What verification have you implemented to justify the word "efficiently" in line 222. Justify the use of an average biocide concentrations to calculate BE. Moreover, why the used samples were not described in the 2.3 section? What is the number of the samples and the representativity? What is the sampling frequency? What is the variability of the measured concentrations? Does the concentration vary in time? Decrease? To prevent the reader's doubts, a part of the explanation from line 349 to 355 could be used in the methodology presentation and the fact that the estimated BE will be compared to the literature.*
Thank you for this comment and your suggestions. We are aware that our approach is limited and only a rough estimate. We chose the word "efficiently" to stress that by only very few samples and little information on the building we obtained realistic estimations on biocide emissions over a two year time period. Sampling is described in 2.3.1 as part of sampling the rain downpipes. We describe concentrations in section 3.2.2. They vary for different events and rather decrease over time which compares to expectations. But the exact temporal evolution of concentrations cannot be determined

based on only four and five point samples of events. We modified this paragraph to clarify the limitation of our approach already in the method section.

In Paragraph 2.5: we added the reference to the sampling and concentration sections:

Since it was situated on top of a flat roof, all storm water including biocide emissions was collected by the rain downpipes (see 2.3.1 and 3.2.2).

Sentences added to 2.5 (as suggested by referee #1):

Note that BE is only a rough estimation with various limitations discussed in 3.2.3. These include applied literature values for initial amount of biocides and paints, no consideration of dry and wet periods or wind driven rain, material aging and limited sampling. We compared the estimated BE with literature values to determine whether estimations are feasible.

Sentence changed in 3.2.3 (as suggested by referee #1):

Regardless of these uncertainties, we arrived at a realistic order of magnitude why we consider our approach promising for an initial estimation of relevant biocide sources by a limited number of samples.

*Paragraph 3.1: I am wondering if 4 sampled events are sufficient to assess the variability of the concentrations in the swale, especially since only one PNEC exceedance is observed to justify the continuation of the study. Why don't you continue to sample the swale in the second part of the study?*

See our answer to paragraph 2.1 above:

Thanks for this comment. On a first glance, it really seemed obvious to continue swale sampling also in 2019-2020. However, as shown in Figure 4, biocide concentrations in the swale are highly variable and depended inter alia on event magnitude. Hence, we did not expect new findings from a renewed sampling campaign here. Instead, our objective was to concentrate on biocide sources and thereby limit the number of samples in an efficient campaign. We stressed this point in the updated manuscript by adding a sentence to section 3.1:

Due to dependencies on rain event magnitudes and thus high variability of detected concentrations in the swale, we did not expect new findings from a renewed sampling campaign in 2019-2020 and thus focused on sources of biocides in the next step.

*Line 298: You said that you sampled an additional pipe (R4-2) because R4-1 exceeded R1 and R2 by an order of magnitude but you have no result for R1 and R2 before the first sampling of R4? I don't understand.*

Your observation is correct, thank you for finding this contradiction. We first sampled one pipe at one building (R4-1) and found concentrations that exceeded expected concentrations, because we did not expect roof areas as a biocide source. During the next event, we decided to sample another pipe at the same building (R4-2) to make sure there was no contamination in the first pipe. Additionally, we decided to sample one pipe at another house (R2). To confirm the low concentrations we sampled at a third building (R1). For comparison purposes we then sampled multiple events at all pipes. We made this clear in the updated manuscript and revised the corresponding paragraph at 3.2.2:

Concentrations in pipe R4-P1 were high, as we did not expect roof areas as a biocide source. We sampled an additional pipe at the same building (R4-P2) to exclude potential contamination in the first pipe. Both R4 pipes showed concentrations of a similar magnitude. For comparison, we decided to sample one pipe at another house (R2). To confirm the low concentrations we sampled at a third building (R1). We then sampled multiple events at all pipes.

*Figure 8: Precise if it is mean or median values in the legend*

Shown are mean values. We added this to the figure description.

Figure description changed to:

Figure 8: Schematic view of roof area of House 4 with sampling spots. Results of sampling for (a) roof materials include roof balustrade, railings, roofing foil, roof access, roof cladding, elevator shaft foil and grass foil, (b) newly painted roof facade and (c) the old roof facade. Bars show mean, lines show minimum and maximum of sample category.

*Line 320 and following: It is not clear if the difference of concentrations is due to the new paint or to the exposition.*

We changed this sentence to make it clear that the different concentrations are due to the new paint.

> We added a sentence and changed the following sentence to ensure consistent meaning.
>
> Thus, high concentrations of detected biocides are likely due to new paint. The western exposure suggests a higher emission rate of biocides due to a higher amount of wind driven rain at the weather side (Vega-Garcia et al., 2020).

*Line 363/364: you explain that OIT was not detected due to its degradation in soil but for S9 (surface water pipe), water is not percolated through the soil? How do you explain to not found OIT in S9 samples?*

Concentrations of OIT at the facades were very low, i.e. 0.9 - 2.3 ng L$^{-1}$. Hence, we did not expect to find OIT in the pipes. We added this point to the manuscript.

> OIT concentrations at the facades were very low (Fig. 6), we thus did not expect to find OIT in the surface runoff pipe S9.

*Line 413: you have to qualify this sentence because your method gave a rough and short term estimate (even if interesting). the comparison to the literature is only informative and does not bring proof of the accuracy of the evaluation, the initial stock of biocides not being necessarily the same.*

Thank you for pointing this out. We clarified this statement. Samples at the rain down pipes just confirmed a continuous biocide leaching from the flat roof. Based on available measurements we did a rough estimation of the total long-term biocide leaching and a comparison of the obtained estimation with literature values to check, if we arrived at a realistic order of magnitude. We clarified this both in the method and in the result section, see also our response to the comment on paragraph 2.5 above.

> In 3.2.3 (results) the following sentences were changed:
>
> The location of the roof facade on top of a flat roof permitted a rough estimation of long-term biocide leaching with only a minimum number of samples at the rain downpipes. The obtained results were in the same order of magnitude as previous studies of artificial walls under natural weather conditions (Burkhardt et al., 2012; Bollmann et al., 2016; Bollmann et al., 2017b).
>
> See also our comment to paragraph 2.5 above:
>
> In Paragraph 2.5, we added the reference to the sampling and concentration sections:
>
> Since it was situated on top of a flat roof, all storm water including biocide emissions was collected by the rain downpipes (see 2.3.1 and 3.2.2).
>
> Sentences added to 2.5 (as suggested by referee #1):
>
> Note that BE is only a rough estimation with various limitations discussed in 3.2.3. These include applied literature values for initial amount of biocides and paints, no consideration of dry and wet periods or wind driven rain, material aging and limited sampling. We compared the estimated BE with literature values to determine whether estimations are feasible.
>
> Sentence changed in 3.2.3 (as suggested by referee #1):
>
> Regardless of these uncertainties, we arrived at a realistic order of magnitude why we consider our approach promising for an initial estimation of relevant biocide sources by a limited number of samples.

*Technical corrections*
*Figure 1: Step 2: perhaps precise "Elution and leaching test experiments"* - corrected.

> See corrected Fig. above, referee #1.

*Line 97: two -s at "sselected"* - corrected.

*Line 114: The capital letter at "Area" is unnecessary* – corrected:

> 2.2 Study area and sampling sites

Roof areas are of diverse uses such as roof top terraces and solar panels, both in combination with extensive green roofs.

*Line 134: I find that "surface water pipe" do not describe well the type of water sampled. It looks like surface water that has been sampled. Perhaps "surface runoff pipe" would be more meaningful*
Thank you for your suggestion, we changed the term accordingly.
Changed throughout the manuscript (8x) and in Figure 2 and Figure 3.

*Figure 5: you could cut the ordinate-axis to better present the lowest concentrations.*
Thank you for your suggestion, we changed the figure accordingly.
Figure 5, changed:

[Figure]

*Line 246: space is missing between below and 4* – corrected:
Please see following comment for entire sentence.

*Line 245, 246 and 247: the sentence is not simple to understand*
We changed this sentence to make it clearer:
In the same study, diuron showed maximum concentrations of up to 0.6 µg L$^{-1}$ and OIT up to 60 ng L$^{-1}$. Gasperi et al., 2013 analyzed storm water at the outlet of three catchments in Paris, Nantes and near Lyon. Mean concentrations were 2 µg L$^{-1}$ for diuron and OIT concentrations remained below 4 ng L$^{-1}$.

*Line 254: substanceS* – corrected:
Differences in types and concentrations of detected substances between locations and events might be due to different sources (e.g. newly painted facades), different intensities of UV radiation and different precipitation amounts and intensities that affect biocide emissions (Paijens et al., 2019).

*Line 296: perhaps add a coma after "in all rain downpipes"* – corrected:
In all rain downpipes, biocides were detected indicating the application of biocides on the flat roofs. However, concentrations of biocides and TPs largely differed between pipes and houses (Fig.7).

*Figure 7 (d): perhaps precise "non sampled event". The use of one single scale is understandable but does not allow to read the concentrations for R1 and R2*
We added non-sampled events in the explanation of the figure.
Figure Description changed to:
Figure 7: Sampled events in rainwater downpipes at three houses. Number after "R" refers to the number of the house. Non-sampled events shown without date.

*Line 321: due "to"?* – corrected:

The part's western exposure suggests a higher emission rate of biocides due to a higher amount of wind driven rain at the weather side (Vega-Garcia et al., 2020).

*Line 579: "TEXTE"?* – changed:
Tietje, O., Burkhardt, M., Rohr, M., Borho, N., and Schoknecht, U.: Emissions- und Übertragungsfunktionen für die Modellierung der Auslaugung von Bauprodukten, UBA, 28, 59 pp., 2018.

**References**

Bollmann, U. E., Fernández-Calviño, D., Brandt, K. K., Storgaard, M. S., Sanderson, H., and Bester, K.: Biocide Runoff from Building Facades: Degradation Kinetics in Soil, Environmental science & technology, 51, 3694–3702, doi:10.1021/acs.est.6b05512, 2017.

Bollmann, U. E., Minelgaite, G., Schlüsener, M., Ternes, T., Vollertsen, J., and Bester, K.: Leaching of Terbutryn and Its Photodegradation Products from Artificial Walls under Natural Weather Conditions, Environmental science & technology, 50, 4289–4295, doi:10.1021/acs.est.5b05825, 2016.

Bollmann, U. E., Vollertsen, J., Carmeliet, J., and Bester, K.: Dynamics of biocide emissions from buildings in a suburban stormwater catchment - concentrations, mass loads and emission processes, Water research, 56, 66–76, doi:10.1016/j.watres.2014.02.033, 2014.

Burkhardt, M., Zuleeg, S., Vonbank, R., Schmid, P., Hean, S., Lamani, X., Bester, K., and Boller, M.: Leaching of additives from construction materials to urban storm water runoff, Water Science and Technology, 63, 1974–1982, doi:10.2166/wst.2011.128, 2011.

Hensen, B., Lange, J., Jackisch, N., Zieger, F., Olsson, O., and Kümmerer, K.: Entry of biocides and their transformation products into groundwater via urban stormwater infiltration systems, Water research, 144, 413–423, doi:10.1016/j.watres.2018.07.046, 2018.

Paijens, C., Bressy, A., Frère, B., Tedoldi, D., Mailler, R., Rocher, V., Neveu, P., and Moilleron, R.: Urban pathways of biocides towards surface waters during dry and wet weathers: Assessment at the Paris conurbation scale, Journal of hazardous materials, 402, 123765, doi:10.1016/j.jhazmat.2020.123765, 2021.

Sauer, F.: Microbicides in Coatings, 143 pp., 2017.

**Answer to Editor Decision**

Thank you for editing our manuscript. Regarding your questions to our answers, we would like to answer this here:

*Question by Christian Stamm: I had one question to an answer you provided. In response to Rev. 3, you listed the analytical recoveries. While the values were ok for most compounds, it was rather low for terbuthylazine-2-hydroxy despite high spiking concentrations. How did you consider that during data analysis?*

Thank you for your question. We are aware that recovery is lower and LOD is higher for Terbuthylazin-2-hydroxy compared to the other investigated compounds. For our analysis we did not take into account these lower recovery rates. For most parts of our analysis a qualitative description of substances was more important than a quantitative description.

---

## Author Response (AR2)

Dear Christian Stamm,

thank you for your comment on the revised version of our manuscript. Please find our responses below.

*L. 140 – 147: Unless there is a clear reason not to do so, please provide the exact information on buildings and facades for all buildings in the SI.*
See line 132: We added the following table in the Appendix and added a reference in L133.

Table A1: Footprint, height and facade area of buildings.

| House No | Footprint size [m²] | Height [m] | Approximate facade area covered by paints and renders [m²] |
|---|---|---|---|
| 1 | 624 | 13 | 634 |
| 2 | 559 | 13 | 577 |
| 3 | 468 | 13 | 525 |
| 4 | 446 | 13 | 484 |
| 5 | 364 | 13 | 426 |
| 6 | 299 | 13 | 369 |
| 7 | 257 | 13 | 343 |
| 8 | 214 | 13 | 296 |

*L. 176: I suggest to write "… are not necessarily representative of the entire rain event." (You don't know whether they were representative or not. Therefore, you cannot make a positive statement that they weren't.)*
See line 164: Thank you for your suggestion. We changed the sentence accordingly.

*L. 196 (figure caption): I suggest to write "Colors correspond to the steps of the experimental procedure and sampling sites."*
See line 182: Thank you for your suggestion. We changed the figure caption accordingly.

*L. 239 – 241: Please add a sentence explaining why the low recovery was not an issue in your case. The low values should not go uncommented.*
See line 229: We added the following explanation:
Terbuthylazin-2-hydroxy and terbumeton had lower recovery rates compared to the other compounds. Hence, the results of the two TPs should be treated with more caution and values might be underestimated. If positive results are obtained despite the mediocre recovery rate, insights into the fate of both TPs can already be obtained.

Please note, we changed the colors of figures 5, 6, 7, 8, 9 and 10 to clarify visual differences of TPs of Terbutryn.

*L: 333: I am not convinced by the water solubility explanation. Even for the parent compounds the solubility cannot be limiting being in the mg/L range (Tab. 3). Please reconsider the argument.*
See line 313: Thank you for pointing this out. We agree with your argument and we deleted this sentence: "Higher water solubility of most TPs compared to biocides might lead to more wash-off of TPs and thus easier detection of TPs." We deleted "Additionally," in the following sentence.